# UDP-glucuronate metabolism controls RIPK1-driven liver damage in nonalcoholic steatohepatitis

Tao Zhang [1,2,3,10], Na Zhang[4,5,10], Jing Xing[6,10], Shuhua Zhang[1,2,10], Yulu Chen[4,10], Daichao Xu [4,7,8] & Jinyang Gu [1,2,9] ✉

Hepatocyte apoptosis plays an essential role in the progression of nonalcoholic steatohepatitis (NASH). However, the molecular mechanisms underlying hepatocyte apoptosis remain unclear. Here, we identify UDP-glucose 6-dehydrogenase (UGDH) as a suppressor of NASH-associated liver damage by inhibiting RIPK1 kinase-dependent hepatocyte apoptosis. UGDH is progressively reduced in proportion to NASH severity. UGDH absence from hepatocytes hastens the development of liver damage in male mice with NASH, which is suppressed by RIPK1 kinase-dead knockin mutation. Mechanistically, UGDH suppresses RIPK1 by converting UDP-glucose to UDP-glucuronate, the latter directly binds to the kinase domain of RIPK1 and inhibits its activation. Recovering UDP-glucuronate levels, even after the onset of NASH, improved liver damage. Our findings reveal a role for UGDH and UDP-glucuronate in NASH pathogenesis and uncover a mechanism by which UDP-glucuronate controls hepatocyte apoptosis by targeting RIPK1 kinase, and suggest UDP-glucuronate metabolism as a feasible target for more specific treatment of NASH-associated liver damage.

Nonalcoholic fatty liver disease (NAFLD) and a more severe form of this progressive liver disease, nonalcoholic steatohepatitis (NASH), is becoming a significant public health concern worldwide and currently has no approved therapies[1]. The clinical manifestations of NAFLD only become problematic at the stage of NASH, when hepatocellular death, inflammation and varying degrees of fibrosis are superimposed on the initial steatosis, which increases the risk for the development of end-stage liver disease, cirrhosis, and hepatocellular carcinoma (HCC)[2]. Hepatocellular death is a cardinal feature of NASH pathogenesis, which causes progressive liver damage, fibrosis and cirrhosis[3,4]. In patients

with NASH, fibrosis is the main determinant of mortality[5–7]. Therefore, preventing liver damage and fibrosis, which may be achieved by improving hepatocyte death, are major goals of NASH therapy[8,9]. However, the molecular mechanisms underlying hepatocyte death in NASH remain unclear.

In humans, hepatocyte apoptosis is the predominant cell death type in NAFLD, which is greater in patients with NASH than in those with simple steatosis, and excessive hepatocyte apoptosis is a pathologic hallmark of NASH[10,11]. Moreover, the magnitude of hepatocyte apoptosis correlates with degree of inflammation and stage of

[1]Center for Liver Transplantation, Union Hospital, Tongji Medical College, Huazhong University of Science and Technology, Wuhan 430022, China. [2]Key Laboratory of Organ Transplantation, Ministry of Education; NHC Key Laboratory of Organ Transplantation; Key Laboratory of Organ Transplantation, Chinese Academy of Medical Sciences, Wuhan 430022, China. [3]Division of Endocrinology, Boston Children's Hospital, Harvard Medical School, Boston, MA 02115, USA. [4]Interdisciplinary Research Center on Biology and Chemistry, Shanghai Institute of Organic Chemistry, Chinese Academy of Sciences, Shanghai 201210, China. [5]University of Chinese Academy of Sciences, Beijing 101408, China. [6]Lingang Laboratory, Shanghai 200031, China. [7]Shanghai Key Laboratory of Aging Studies, Shanghai 2012010, China. [8]State Key Laboratory of Chemical Biology, Shanghai Institute of Organic Chemistry, Chinese Academy of Sciences, Shanghai 200032, China. [9]Department of Transplantation, Xinhua Hospital Affiliated to Shanghai Jiao Tong University School of Medicine, Shanghai 200092, China. [10]These authors contributed equally: Tao Zhang, Na Zhang, Jin Xing, Shuhua Zhang, Yulu Chen. ✉e-mail: gjynyd@126.com

fibrosis[10]. In hepatocytes, apoptosis is often initiated by extracellular death receptor ligands, such as the tumor necrosis factor (TNF) superfamily, and mediated by caspase-8[12]. Indeed, steatosis induced by long-term feeding of mice with a high-fat diet (HFD) sensitizes hepatocytes to cytokine-induced apoptosis, as may occur during NASH when hepatic leukocytes locally secrete pro-inflammatory cytokines, such as TNFα[13]. TNFα is critically involved in the pathogenesis and disease progression of NASH[9,14]. In TNFα signaling, the serine/threonine kinase RIPK1 functions as a master upstream regulator of both cell survival mediated by its scaffold function and cell death mediated by its kinase function[15]. Activation of RIPK1 kinase promotes much of the deleterious effects, including apoptosis, necroptosis, and inflammation, activated by TNFα in human diseases[16]. RIPK1 activation has also been found in both human patients with NASH and mouse models of NASH. Pharmacological or genetic inhibition of RIPK1 kinase can reduce hepatic inflammation and liver injury in HFD-induced NAFLD[17,18]. However, the molecular mechanism underlying RIPK1 activation in NASH is largely unknown.

Emerging data from human genetics supports a role for metabolic pathways in NAFLD and risk for progression to NASH. For example, mutations in 1-acylglycerol-3-phosphate O-acyltransferase (PNPLA3), and hydroxysteroid dehydrogenase 13 (HSD17B13), have been shown to increase risk for NASH[19,20]. Mutations in genes encoding proteins that regulate glucose metabolism, such as glucokinase regulator (GCKR), have also been linked to NASH risk[21]. UDP-glucose 6-dehydrogenase (UGDH) is a key cytosolic enzyme in the uronic acid pathway, and converts uridine diphosphate (UDP)-glucose (UDP-Glc) to UDP-glucuronate (UDP-GlcA) through the concomitant reduction of $NAD^+$ into NADH[22,23]. UDP-GlcA is not only needed for detoxification via glucuronidation, but is also further converted to UDP-xylose by UDP-xylose synthase (UXS1)-catalyzed decarboxylation reaction, which is an obligate precursor for the synthesis of glycosaminoglycans, and therefore an important component of proteoglycans of the extracellular matrix[24,25]. UGDH is a key player for the production of extracellular matrix components that are essential for human brain development, and its mutations are frequent cause of recessive epileptic encephalopathy[26]. In addition, dysregulation of UGDH is central in various human cancer progression and metastasis[27–30]. Although UGDH functions in certain systems, its pathophysiological role has not been well defined. Moreover, whether and how UGDH as well as UDP-glucuronate metabolism contribute to NASH remain unexplored.

In this study, we identify UGDH as a suppressor of hepatocyte apoptosis induced by TNFα. We find that deficiency of UGDH sensitizes hepatocytes to apoptosis in a RIPK1 kinase-dependent manner following TNFα stimulation. UGDH and UDP-GlcA are markedly downregulated in the livers of individuals with NAFLD or NASH, and UGDH absence from mouse hepatocytes promotes RIPK1 activation and hastens the development of NASH-associated liver damage and fibrosis by exaggerating hepatocyte apoptosis, which is suppressed by RIPK1 kinase-dead D138N knockin mutation. We demonstrate that UGDH suppresses RIPK1 activation by converting UDP-Glc to UDP-GlcA, which promotes the UDP-GlcA-mediated inhibition of RIPK1 kinase and therefore limits hepatocyte apoptosis induced by TNFα. We show that UDP-GlcA, but not UDP-Glc, directly binds to the kinase domain of RIPK1 and inhibits its activation. Moreover, recovering UDP-GlcA levels by adeno-associated virus-mediated overexpression of UGDH or knockdown of UXS1 in the liver in mice models of NASH substantially reduced hepatocyte apoptosis and fibrosis in this disease. Notably, we also show the therapeutic potential of UDP-GlcA in NASH-associated liver damage in mice models. Collectively, our data reveal a previously unappreciated role for UGDH and UDP-GlcA in NASH pathogenesis by controlling RIPK1 activation and hepatocyte apoptosis, and suggest UDP-GlcA metabolism as a feasible target for more specific treatment of NASH-associated liver damage.

## Results

### Identification of UGDH as a suppressor of hepatocyte apoptosis induced by TNFα

Whether metabolic pathways contribute to hepatocyte apoptosis remains unknown, we sought to identify the metabolic pathway that regulates TNFα-induced hepatocyte apoptosis stimulated with TNFα and the translation inhibitor cycloheximide (CHX), a well-established paradigm by which apoptosis is induced and mediated by caspase-8. We used short interfering RNAs (siRNAs) to knock down 114 metabolic enzymes or *Casp8* (as the positive control) in primary hepatocytes to identify the metabolic enzyme(s) that might regulate hepatocyte apoptosis. Among these enzymes, *Ugdh* knockdown greatly sensitized hepatocytes to TNFα-induced apoptosis (Fig. 1a, b). UGDH knockdown substantially enhanced apoptosis as shown by increased levels of the cleaved caspase-8 (CC8) and cleaved caspase-3 (CC3) (Fig. 1c). The Human Protein Atlas (HPA) RNA-sequencing data showed that *UGDH* mRNA is mainly enriched in the liver and highly expressed by hepatocytes (Supplementary Fig. 1a, b). Consistent to its mRNA expression levels in human, UGDH is also highly expressed in mouse liver and enriched in hepatocytes, and rarely expressed in Kupffer cells and liver endothelial cells (Supplementary Fig. 1c, d). We then generated hepatocyte-specific UGDH-knockout mice by crossing *Ugdh*[f/f] mice with *Albumin-Cre* (*Alb-Cre*) Tg mice to elucidate the function of UGDH in hepatocyte apoptosis. The efficient deletion of UGDH in hepatocytes was confirmed by immunoblotting analysis of whole liver and isolated primary hepatocytes (Supplementary Fig. 1e, f). Consistently, UGDH knockout greatly sensitized hepatocytes to TNFα-induced apoptosis with activation of caspase-8 and caspase-3 (Fig. 1d, e).

### UGDH deficiency licenses TNFα-induced RIPK1-dependent apoptosis in hepatocytes

Both wild-type (WT) and UGDH knockout hepatocytes treated with TNFα/CHX underwent apoptosis, which can be blocked by the pan-caspase inhibitor zVAD.fmk (zVAD) (Fig. 1f). In contrast, necroptosis, ferroptosis, and pyroptosis are not involved in the UGDH deficiency-induced exaggerated cell death phenotype in hepatocytes, as treatments with inhibitors of these cell death modalities in UGDH knockout hepatocytes have no effect on cell death induced by TNFα/CHX (Fig. 1f). The treatment of TNFα can induce two apoptotic modalities, including RIPK1 kinase-independent apoptosis (RIA) and alternatively RIPK1 kinase-dependent apoptosis (RDA) in certain conditions[15]. RIA can be converted into RDA when RIPK1 is prone to activation[31,32]. TNFα/CHX usually induce RIA, but not RDA in normal conditions. Consistently, WT hepatocytes treated with TNFα/CHX also underwent RIA, as RIPK1 kinase inhibitor Nec-1s[33] cannot protect against apoptosis of WT hepatocytes induced by TNFα/CHX (Fig. 1f, g). Interestingly, we found that the enhanced apoptosis induced by TNFα/CHX in UGDH-deficient hepatocytes was blunted by Nec-1s (Fig. 1f, g). Consistently, co-treatment with TNFα/CHX and Nec-1s reduced the levels of activated caspase-8 and caspase-3 in UGDH knockout hepatocytes, but not in WT hepatocytes (Fig. 1h). These results suggest that RIPK1 might be activated in the absence of UGDH in response to TNFα. In line with this notion, we detected substantial amount of activated RIPK1 as determined by its activation biomarker p-S166 RIPK1[33,34], in UGDH-deficient hepatocytes treated with TNFα/CHX, but not that in WT hepatocytes (Fig. 1h). We subsequently characterized the interaction of RIPK1 and caspase-8 with FADD to form complex II, a key downstream event that promotes activation of caspases-8 in TNFα-induced apoptosis[35]. UGDH knockout led to an increased formation of complex II as compared to WT hepatocytes, and interaction of activated RIPK1 and activated caspase-8 with FADD induced by TNFα/CHX in UGDH-deficient hepatocytes was blocked by Nec-1s (Fig. 1i). We then treated primary hepatocytes with TNFα alone, a more relevant pathological condition, to mimic inflammation. We found that UGDH knockout also greatly sensitizes hepatocytes to apoptosis induced by TNFα alone (Fig. 1j).

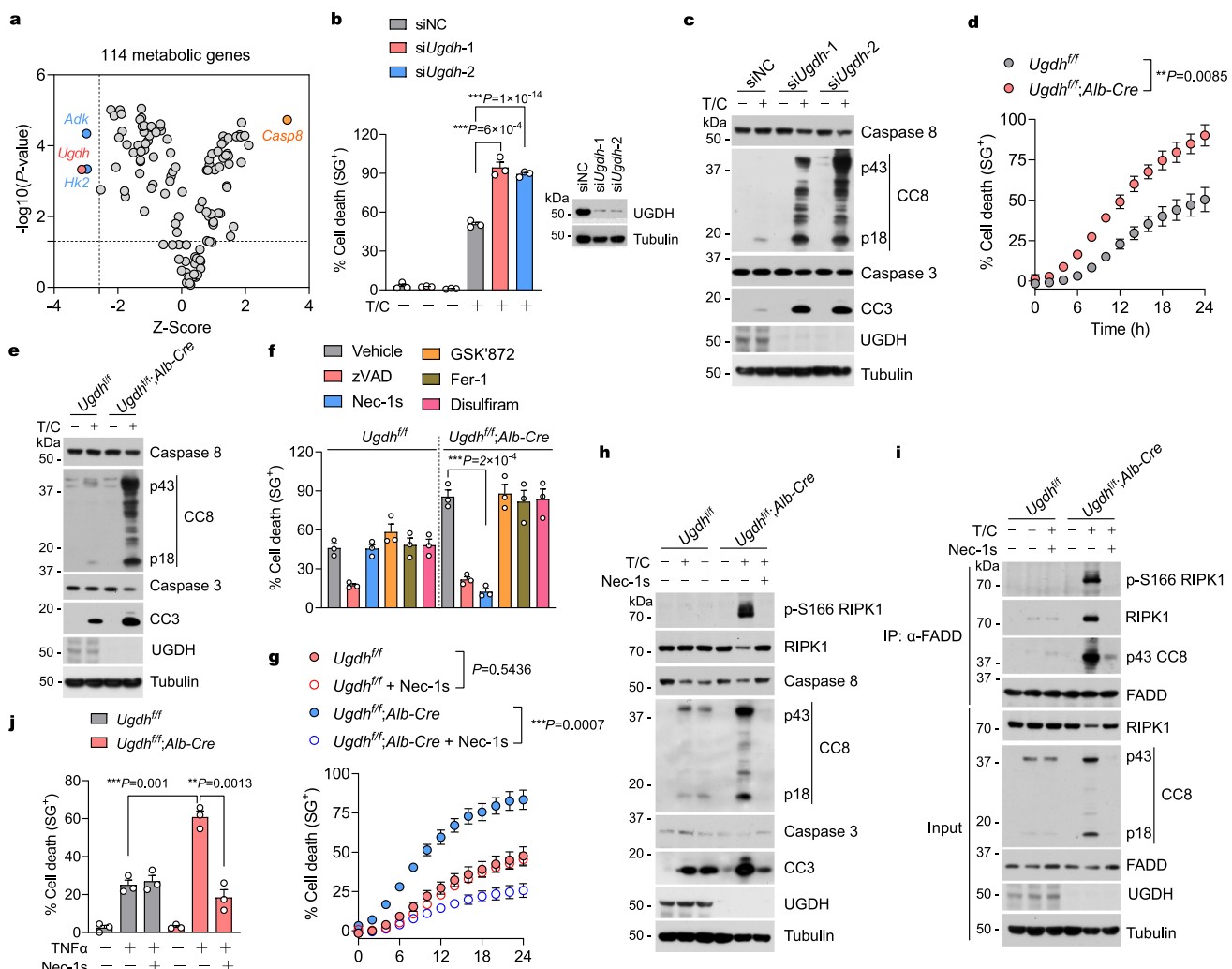

**Fig. 1 | Identification of UGDH as a suppressor of hepatocyte apoptosis mediated by RIPK1. a** Primary hepatocytes were transfected with siRNA pools targeting 114 metabolic enzymes for 48 h followed by treatment with TNFα (T, 10 ng/ml) and cycloheximide (CHX, 1 μM) and cultured for an additional 24 h to induce cell death. The screen hits were selected with cut-offs set at Z-score < −2.576 and P-value < 0.05. Yellow, CASP8 (positive control). Red, UGDH. Blue, ADK and HK2. Primary hepatocytes were transfected with siRNAs targeting UGDH, or nontargeting control (NC), for 48 h. The cells were then treated with CHX (1 μM)/TNFα (10 ng/ml) for 24 h (**b**) or 12 h (**c**). Cell death was measured by SytoxGreen positivity assay (**b**). The levels of cleaved caspase-8 (CC8) and cleaved caspase-3 (CC3) were determined by immunoblotting (**c**). Primary hepatocytes were treated with CHX (1 μM)/TNFα (10 ng/ml) for indicated time (**d**) or 12 h (**e**) or 24 (**f**) in the presence or absence of zVAD.fmk (zVAD, 10 μM), Nec-1s (10 μM), GSK'872 (1 μM), ferrostatin-1 (Fer-1, 10 μM), and disulfiram (50 μM). Cell death was measured by SytoxGreen positivity assay (**d, f**). The levels of CC8 and CC3 were determined by immunoblotting (**e**). Primary hepatocytes were treated with CHX (1 μM)/TNFα (10 ng/ml) for indicated time (**g**) or 12 h (**h, i**) in the presence or absence of Nec-1s (10 μM). Cell death was measured by SytoxGreen positivity assay (**g, i**). The levels of p-S166 RIPK1, CC8 and CC3 were determined by immunoblotting (**h**). The complex II was isolated by immunoprecipitation of FADD, RIPK1 and caspase-8 binding were revealed by immunoblotting (**i**). **j** Primary hepatocytes were treated with TNFα (10 ng/ml) for 24 h in the presence or absence of Nec-1s (10 μM). Cell death was measured by SytoxGreen positivity assay. Data are represented as mean ± s.e.m. of n = 3 independent experiments (**b, d, f, g, j**). Similar results were obtained from n = 3 independent experiments (**c, e, h, i**). Unpaired two-tailed t-test (**a, f**). One-way ANOVA, post hoc Dunnett's test (**b, j**). Two-way ANOVA (**d, g**). Source data are provided as a Source Data file.

The increased sensitivity of UGDH-deficient hepatocytes to TNFα alone-induced apoptosis was also suppressed by RIPK1 kinase inhibitor Nec-1s (Fig. 1j). Thus, UGDH deficiency not only sensitizes hepatocytes to apoptosis induced by TNFα, but also promotes the activation of RIPK1 and converts RIA into RDA in response to TNFα.

### Reduced UGDH expression in fatty liver is associated with the severity of NAFLD

Given the important role of excessive TNFα-induced hepatocytes apoptosis in NASH-associated liver damage, we then explored the involvement of UGDH in the progression of NASH. We first examined UGDH expression in the livers of human subjects without steatosis, with simple steatosis and with NASH. We found that steady-state UGDH protein levels were considerably lower in the livers of individuals with simple steatosis or NASH than in the nonsteatotic controls, and the NASH group had markedly lower UGDH expression than the simple steatosis group (Fig. 2a). Notably, UGDH protein levels in the liver were negatively correlated with NASH activity score (NAS) (Fig. 2b). In addition, UGDH protein levels were also negatively correlated with the serum levels of alanine transaminase (ALT) and aspartate aminotransferase (AST), two commonly used biomarkers for liver damage (Fig. 2c). We then examined UGDH expression in the livers of mouse NASH models. We fed mice with a choline-deficient high-fat-diet (CD-HFD) to rapidly induce hepatic steatosis, inflammation, liver damage, and fibrosis, which are characteristics of NASH[36]. CD-HFD

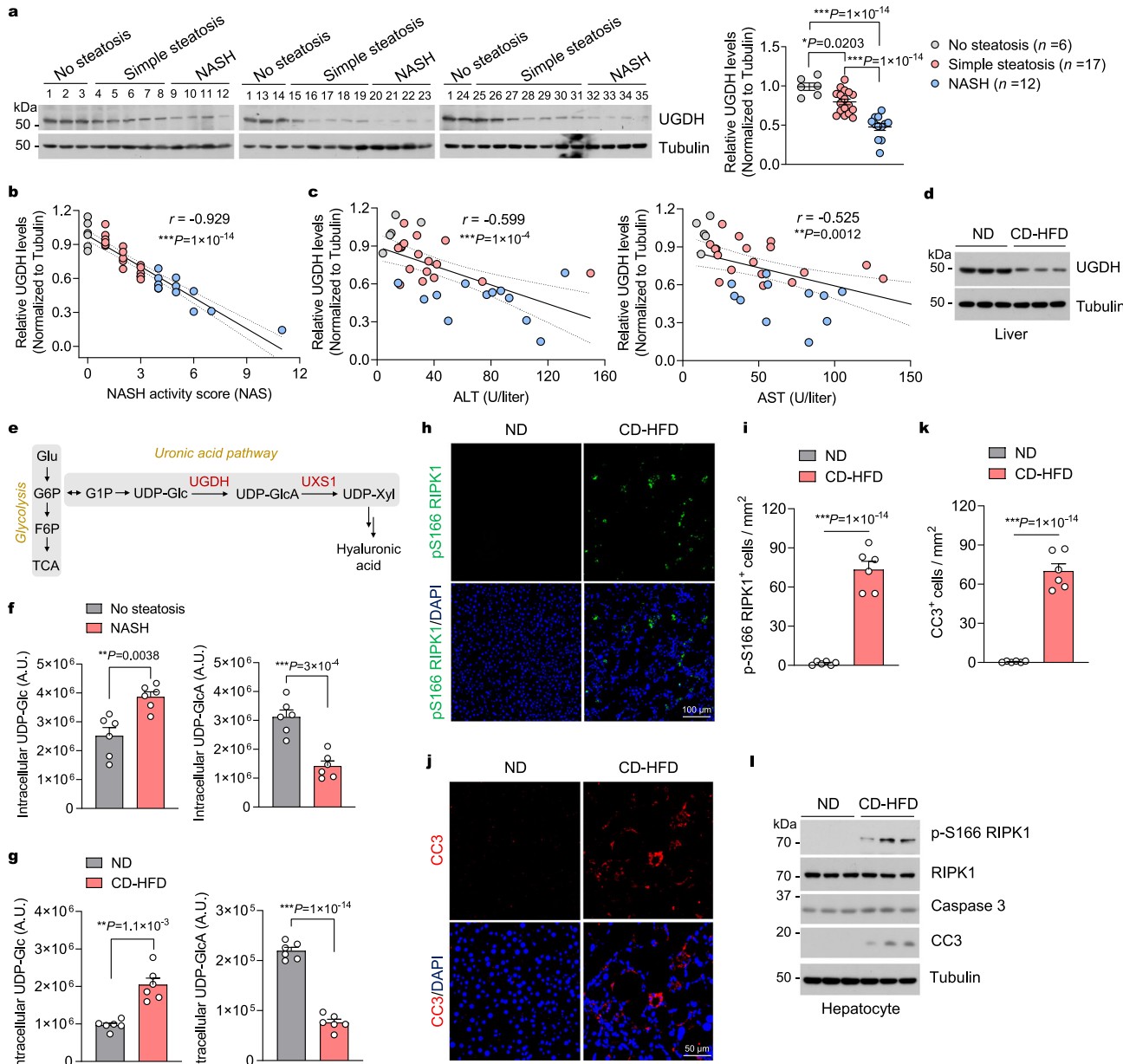

**Fig. 2 | Reduced UGDH expression in fatty liver is associated with the severity of NAFLD. a** Immunoblotting and quantification of UGDH expression in the livers of individuals with non-steatosis ($n = 6$), simple steatosis ($n = 17$) or NASH ($n = 12$). Protein expression was normalized to Tubulin levels and shown as relative values to sample 1. Pearson comparison analyses of the correlation between UGDH protein level (normalized to Tubulin level) and NAS (**b**), serum ALT and AST concentrations (**c**). Spearman's rank correlation coefficient analysis. **d** Representative immuno-blotting analysis of UGDH protein levels in the livers of 20-week-old mice fed ND or CD-HFD for 12 weeks. $n = 3$ mice per group. **e** Schematic diagram of the uronic acid pathway and glycolysis. G6P glucose-6-phosphate, F6P fructose-6-phosphate, G1P glucose-1-phosphate, Glc glucose, GlcA Glucuronate, Xyl xylose. TCA tricarboxylic acid cycle. Intracellular UDP-Glc and UDP-GlcA concentrations in the livers of individuals with non-steatosis ($n = 6$), or NASH ($n = 6$) (**f**), and 20-week-old mice that were fed ND ($n = 6$) or CD-HFD ($n = 6$) for 12 weeks (**g**). **h, i** Immunofluorescence

images of p-S166 RIPK1 of liver sections from 20-week-old mice with indicated genotypes after feeding with CD-HFD for 12 weeks. Representative images out of $n = 6$ mice for each genotype are represented (**h**). Graph depicting numbers of p-S166 RIPK1+ cells on liver sections of indicated genotypes. Each dot represents an individual mouse (**i**). **j, k** Immunofluorescence images of CC3 of liver sections from 20-week-old mice with indicated genotypes after feeding with CD-HFD for 12 weeks. Representative images out of $n = 6$ mice for each genotype are represented (**j**). Graph depicting numbers of CC3+ cells on liver sections of indicated genotypes (**k**). Each dot represents an individual mouse (**k**). **l** Representative immunoblotting analysis of p-S166 RIPK1 and CC3 protein levels in primary hepatocytes isolated from livers of 20-week-old mice fed ND or CD-HFD for 12 weeks. $n = 3$ mice per group. Mean ± s.e.m. (**a, f, g, i, k**). One-way ANOVA, post hoc Dunnett's test (**a**). Unpaired two-tailed $t$-test (**f, g, i, k**). Source data are provided as a Source Data file.

decreased UGDH expression levels in mouse livers as compared to normal chow diet (ND)-fed controls (Fig. 2d and Supplementary Fig. 1g). We also fed mice with the Amylin (AMLN) diet, which is used to mimic human NASH in preclinical studies[8,37]. AMLN also reduced UGDH protein levels in mouse livers (Supplementary Fig. 1h). Although protein levels varied, *Ugdh* mRNA levels remained not significantly changed in livers from mice fed with CD-HFD or

AMLN as compared to ND-fed controls or from human NASH livers as compared to no steatosis controls (Supplementary Fig. 1i). In addition, UGDH levels were not altered in primary hepatocytes after exposure to palmitic acid, which was performed to mimic in vivo NAFLD pathologies (Supplementary Fig. 1j), suggesting that the decline of UGDH expression in livers with NAFLD is not attributed to the accumulation of palmitate per se.

UGDH is the key enzyme in the uronic acid pathway that catalyzes the conversion of UDP-Glc to UDP-GlcA and participates in the biosynthesis of glycosaminoglycans, such as hyaluronic acid (Fig. 2e). Consistent to the reduction of UGDH expression in livers from patients with NASH, the cellular UDP-Glc levels were increased while the UDP-GlcA levels were substantially decreased in the livers of NASH patients as compared to that of nonsteatotic controls (Fig. 2f). Furthermore, the cellular levels of UDP-Glc were also increased and UDP-GlcA were decreased in livers from mice fed with CD-HFD or AMLN (Fig. 2g and Supplementary Fig. 1k). Because UGDH reduction sensitized hepatocytes to TNFα-induced apoptosis by promoting RIPK1 activation, we then examined whether RIPK1 activation can be seen in livers of CD-HFD-fed mice. We detected RIPK1 activation as determined by substantial amount of p-S166 RIPK1 positive cells in liver sections of CD-HFD-fed mice (Fig. 2h, i). Consistently, we detected increased apoptotic cells in livers of CD-HFD-fed mice as assessed by CC3 immunostaining and terminal deoxynucleotidyl transferase-mediated deoxyuridine triphosphate nick end labeling (TUNEL) staining (Fig. 2j, k and Supplementary Fig. 1l). The activation of RIPK1 and apoptosis were further confirmed by immunoblotting of p-S166 RIPK1 and CC3 in primary hepatocytes isolated from CD-HFD-fed mice (Fig. 2l). In contrast, p-S166 RIPK1 and CC3 were not present in Kupffer cells and liver endothelial cells isolated from CD-HFD-fed mice (Supplementary Fig. 1m).

### Hepatocyte-specific UGDH knockout exaggerates liver damage in CD-HFD-induced NASH

We found that hepatocyte-specific UGDH ablation did not affect body weight, liver weight, or triglycerides (TGs) in mice fed normal chow diet (ND) (Supplementary Fig. 2a, b). ND-fed *Ugdh*[f/f];*Alb-Cre* mice had normal serum ALT, AST levels and liver morphology (Supplementary Fig. 2c, d). These results suggest that although UGDH is decreased in NAFLD, loss of UGDH does not induce NAFLD-associated pathologies in unstressed condition. We next investigated the function of UGDH-mediated suppression of apoptosis in the pathogenesis of diet-induced NASH. We fed *Ugdh*[f/f] and *Ugdh*[f/f];*Alb-Cre* mice with CD-HFD to induce NASH-related characteristics. *Ugdh*[f/f];*Alb-Cre* mice were identical to *Ugdh*[f/f] mice with respect to body weight and liver weight after feeding CD-HFD for 12 weeks (Supplementary Fig. 2e). Hematoxylin and eosin (H&E) and oil red O (ORO) staining revealed that the hepatic steatosis and lipid accumulation were not altered in CD-HFD-fed *Ugdh*[f/f];*Alb-Cre* mice compared with that of CD-HFD-fed *Ugdh*[f/f] mice (Supplementary Fig. 2f), which was further confirmed by measuring hepatic concentrations of triglycerides (TG), total cholesterol (TC) and nonesterified fatty acids (NEFA) (Supplementary Fig. 2g). However, we found a significant increase of serum ALT and AST levels, indicating exaggerated liver damage, in CD-HFD-fed *Ugdh*[f/f];*Alb-Cre* mice (Supplementary Fig. 3a). In line with this observation, we found increased liver TUNEL staining in CD-HFD-fed *Ugdh*[f/f];*Alb-Cre* mice, demonstrating that knockout of UGDH substantially increased hepatocellular death (Supplementary Fig. 3b). We then assessed RIPK1 activation and apoptosis in CD-HFD-fed *Ugdh*[f/f];*Alb-Cre* mice. We found that loss of UGDH significantly increased RIPK1 activation and apoptosis as determined by p-S166 RIPK1 and CC3 immunostaining, respectively, in liver sections of CD-HFD-fed *Ugdh*[f/f];*Alb-Cre* mice (Supplementary Fig. 3c, d). Consistently, RIPK1 activation and apoptosis as assessed by immunoblotting were both increased in primary hepatocytes isolated from CD-HFD-fed *Ugdh*[f/f];*Alb-Cre* mice when compared with that from CD-HFD-fed *Ugdh*[f/f] mice (Supplementary Fig. 3e). In addition, we also observed substantial increase in the levels of activated RIPK1 and apoptosis in human NASH livers that showed reduced expression of UGDH (Supplementary Fig. 3f).

In human NASH, hepatocyte apoptosis closely correlates with hepatic inflammation and fibrosis[10], we next investigated the effect of UGDH deficiency-induced excessive hepatocyte apoptosis on liver inflammation and fibrosis. We found that CD-HFD-fed *Ugdh*[f/f];*Alb-Cre* mice showed no changes in liver macrophage infiltration, as evidenced by similar macrophage marker CD45 staining in *Ugdh*[f/f] and *Ugdh*[f/f];*Alb-Cre* mice (Supplementary Fig. 3g). Consistently, UGDH knockout did not affect the expression of macrophage marker *Adgre1* (F4/80), the pro-inflammatory cytokines *Tnf* and *Ilb*, or the chemotactic cytokine *Ccl2* (Supplementary Fig. 3h). Nonetheless, hepatic fibrosis as measured with Masson's trichrome staining (MTS) and sirius red staining was increased in CD-HFD-fed *Ugdh*[f/f];*Alb-Cre* mice (Supplementary Fig. 3i). The increased hepatic fibrosis in CD-HFD-fed *Ugdh*[f/f];*Alb-Cre* mice was further confirmed by increased abundance of hydroxyproline, a major component of fibrillar collagen of all types and used as a diagnostic marker of liver fibrosis (Supplementary Fig. 3j). Consistent with increased fibrosis, UGDH deficiency increased the expression of the fibrosis marker gene *Acta2*, collagen genes *Col1a1* and *Col3a1*, as well as fibrogenic growth factors *Pdgfa*, *Pdgfb* and receptor *Pdgfra* (Supplementary Fig. 3k). *Ugdh*[f/f];*Alb-Cre* mice showed no differences in the expression of *Tgfb*, the major macrophage-derived fibrogenic cytokine (Supplementary Fig. 3k), which is consistent with similar macrophage infiltration in CD-HFD-fed *Ugdh*[f/f] and *Ugdh*[f/f];*Alb-Cre* mice.

### Inhibition of RIPK1 kinase activity improves UGDH deficiency-induced liver damage in NASH

To address the function of RIPK1 activation in contributing to exacerbated hepatocyte apoptosis and liver damage in CD-HFD-fed *Ugdh*[f/f];*Alb-Cre* mice, we next generated *Ugdh*[f/f];*Alb-Cre*;*Ripk1*[D138N/D138N] by crossing *Ugdh*[f/f];*Alb-Cre* mice with RIPK1 kinase-dead knockin *Ripk1*[D138N/D138N] mice[38]. In primary hepatocytes, *Ripk1*[D138N/D138N] cells showed the same sensitivity to TNFα/CHX-induced apoptosis as WT cells (Supplementary Fig. 4a), while *Ugdh*[f/f];*Alb-Cre*;*Ripk1*[D138N/D138N] cells showed resistance to TNFα/CHX-induced apoptosis (Fig. 3a). Consistently, TNFα/CHX-treated *Ugdh*[f/f];*Alb-Cre* hepatocytes but not *Ugdh*[f/f] hepatocytes exhibited RIPK1 activation as determined by its S166 phosphorylation and elevated apoptosis as shown by CC8 and CC3, both of which were inhibited in *Ugdh*[f/f];*Alb-Cre*;*Ripk1*[D138N/D138N] hepatocytes (Fig. 3b). In addition, RIPK1 kinase-dead mutation inhibited exacerbated apoptosis in *Ugdh*[f/f];*Alb-Cre* hepatocytes treated with TNFα alone (Fig. 3c). These results further support the role of RIPK1 activation in driving hepatocyte apoptosis in the absence of UGDH in response to TNFα stimulation.

We next examined the role of RIPK1 activation in contributing to enhanced liver damage in CD-HFD-fed *Ugdh*[f/f];*Alb-Cre* mice. Inactivating RIPK1 in *Ugdh*[f/f];*Alb-Cre*;*Ripk1*[D138N/D138N] mice did not affect body weight and liver weight after feeding CD-HFD (Supplementary Fig. 4b), nor did it affect hepatic steatosis and lipid accumulation as determined by H&E and ORO staining, as well as hepatic TG, TC and NEFA measurement (Supplementary Fig. 4c–e). However, RIPK1 kinase-dead mutation significantly prevented RIPK1 activation and reduced the number of apoptotic cells in CD-HFD-fed *Ugdh*[f/f];*Alb-Cre*;*Ripk1*[D138N/D138N] mice as compared to that of *Ugdh*[f/f];*Alb-Cre* mice (Fig. 3d, e and Supplementary Fig. 4f). The inhibited activation of RIPK1 and apoptosis by RIPK1-D138N was further confirmed by immunoblotting of p-S166 RIPK1 and CC3 in primary hepatocytes isolated from CD-HFD-fed mice (Fig. 3f). RIPK1 inhibition has no effect on liver macrophage infiltration, as evidenced by similar macrophage marker CD45 staining in *Ugdh*[f/f];*Alb-Cre* and *Ugdh*[f/f];*Alb-Cre*;*Ripk1*[D138N/D138N] mice that were fed with CD-HFD for 12 weeks (Supplementary Fig. 4g). In addition, RIPK1 inhibition did not affect the expression of pro-inflammatory cytokines *Tnf*, *Ilb*, and *Il6* or the chemotactic cytokine *Ccl2* in livers of *Ugdh*[f/f];*Alb-Cre*;*Ripk1*[D138N/D138N] mice that were fed with CD-HFD for 12 weeks as compared to *Ugdh*[f/f];*Alb-Cre* mice (Supplementary Fig. 4h). Nonetheless, inactivating RIPK1 significantly improved liver damage of *Ugdh*[f/f];*Alb-Cre* mice as determined by serum ALT and AST levels (Fig. 3g). Inhibition of RIPK1

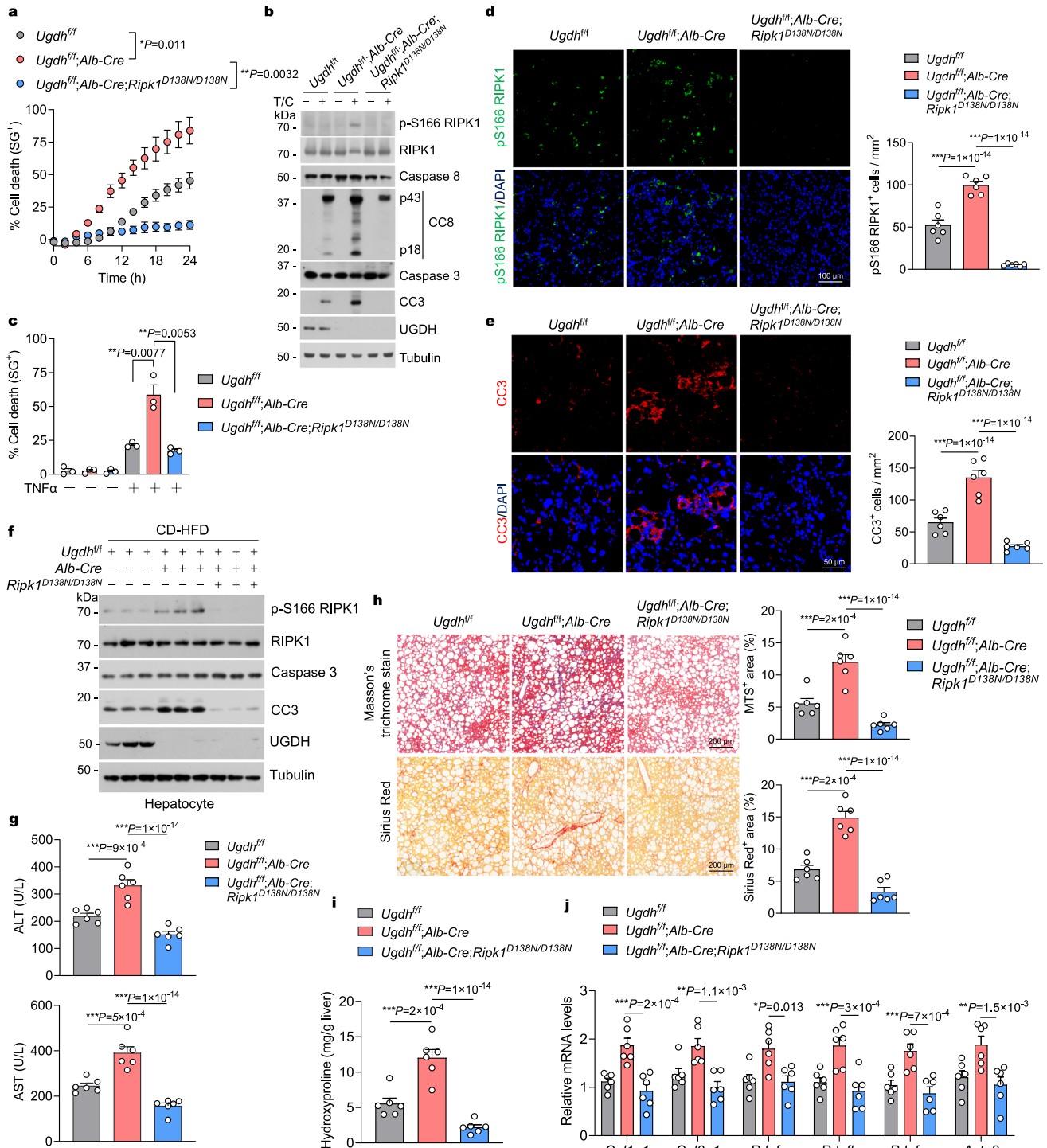

**Fig. 3 | UGDH deficiency increases RIPK1 activation to promote liver damage in NASH.** Primary hepatocytes were treated with CHX (1 μM)/TNFα (10 ng/ml) for indicated time (**a**) or 12 h (**b**). Cell death was measured by SytoxGreen positivity assay. The levels of p-S166 RIPK1, CC8 and CC3 were determined by immuno-blotting (**b**). **c**, Primary hepatocytes were treated with TNFα (10 ng/ml) for 24 h. Cell death was measured by SytoxGreen positivity assay. **d, e,** Immunofluorescence images of p-S166 RIPK1 (**d**) and CC3 (**e**) of liver sections from 16-week-old mice with indicated genotypes after feeding with CD-HFD for 8 weeks. Representative images out of n = 6 mice for each genotype are represented. Graph depicting numbers of p-S166 RIPK1+ (**d**) and CC3+ (**e**) cells on liver sections of indicated genotypes. Each dot represents an individual mouse. **f** Representative immunoblotting analysis of p-S166 RIPK1 and CC3 protein levels in primary hepatocytes isolated from livers of 16-week-old mice with indicated genotypes after feeding with CD-HFD for 8 weeks.

n = 3 mice per group. **g** Serum levels of ALT and AST of 16-week-old mice with indicated genotypes after feeding with CD-HFD for 8 weeks. n = 6 mice for each genotype. Each dot represents an individual mouse. **h** Representative images of Masson's trichrome (MTS) and Sirius red stained liver sections of 16-week-old mice with indicated genotypes after feeding with CD-HFD for 8 weeks. Graph depicting percentage of MTS+ and Sirius red+ area on liver sections of indicated genotypes. n = 6 mice for each genotype. Each dot represents an individual mouse. Liver hydroxyproline (**i**) and mRNA expression of fibrogenic parameters (**j**) in livers of 16-week-old mice with indicated genotypes after feeding with CD-HFD for 8 weeks. n = 6 mice for each genotype. Each dot represents an individual mouse. n = 3 independent experiments (**a–c**). Mean ± s.e.m. (**a, c, d, e, g–j**). Two-way ANOVA (**a**). One-way ANOVA, post hoc Dunnett's test (**c–e, g–i**). Two-way ANOVA, post hoc Bonferroni's test (**j**). Source data are provided as a Source Data file.

also attenuated hepatic fibrosis and significantly reduced the levels of hydroxyproline (Fig. 3h, i). Consistently, RIPK1-D138N mutation also significantly inhibited the expression of fibrotic genes, including *Col1a1*, *Col3a1*, *Pdgfa*, *Pdgfb*, *Pdgfra* and *Acta2* (Fig. 3j).

RIPK1 activation has been shown to promote hepatic steatosis and hepatic inflammation in a conventional high-fat diet (HFD)-induced NAFLD model[17,18]. However, in CD-HFD-fed UGDH-deficient mice, suppression of RIPK1 kinase was unable to reduce hepatic steatosis and inflammation. Since HFD-induced NAFLD model displays less severe pathological outcomes than that of CD-HFD and human NASH, particularly the fibrogenesis process[39], we further investigated whether UGDH-mediated RIPK1 suppression plays a role in the pathogenesis of HFD-induced NAFLD by feeding mice with a conventional HFD for 16 weeks to induce moderate hepatic steatosis and inflammation. To this end, we first characterized the expression levels of UGDH in HFD-fed livers. Unlike CD-HFD, HFD had minor effect on UGDH levels (Supplementary Fig. 5a). Nonetheless, UGDH deficiency also promoted RIPK1 activation in HFD-fed livers (Supplementary Fig. 5b). We also observed a significant increase of serum levels of ALT and AST, suggesting exaggerated liver damage, in HFD-fed UGDH-deficient mice compared with that of control mice, which is markedly reduced by RIPK1-D138N mutation (Supplementary Fig. 5c). In keeping with this observation, HFD-fed UGDH-deficient livers showed increased number of TUNEL-positive cells, which can be prevented by inhibition of RIPK1 kinase (Supplementary Fig. 5d). In addition to a propensity for liver damage, HFD-fed UGDH-deficient mice, in comparison to *Ugdh*[f/f] mice on the same diet, also exhibited greater levels of lipid accumulation and steatosis in the liver, as indicated by H&E and Oil red O staining (Supplementary Fig. 5e), as well as by levels of hepatic TG and TC (Supplementary Fig. 5f). Importantly, the hepatic steatosis and lipid accumulation in the liver of *Ugdh*[f/f]*;Alb-Cre;Ripk1*[D138N/D138N] mice were considerably lower when compared to that of *Ugdh*[f/f]*;Alb-Cre* mice (Supplementary Fig. 5e, f). Moreover, hepatocyte-specific UGDH ablation led to increased mRNA levels of multiple pro-inflammatory markers in mouse liver after 16-week HFD feeding (Supplementary Fig. 5g). Notably, the hepatic expression of those pro-inflammatory markers in *Ugdh*[f/f]*;Alb-Cre;Ripk1*[D138N/D138N] mice were also significantly lower than those in the *Ugdh*[f/f]*;Alb-Cre* mice (Supplementary Fig. 5g). Those results suggested that in a conventional HFD model, which stimulates less severe NAFLD pathology than CD-HFD, UGDH-mediated RIPK1 suppression controls not only hepatocytes cell death and liver damage but also hepatic steatosis and inflammation.

### Restoring UGDH expression improves NASH-associated liver damage

We next evaluated the potential therapeutic effect of restoring UGDH expression against NASH-associated liver damage in mice. We fed mice with CD-HFD for 4 weeks to establish NASH and then intravenously injected mice with adeno-associated virus 8 (AAV8), a liver-targeted therapeutic gene vector that has high transduction efficiency in liver cells[40], to overexpress UGDH (AAV-UGDH) in the livers of mice while continuing CD-HFD for additional 8 weeks (Supplementary Fig. 5h). The empty vector was used as a control (AAV-Ctrl). Exogenous UGDH expression was validated through immunoblotting analysis using liver samples (Fig. 4a). By 8 weeks following treatment, UGDH overexpression increased cellular levels of UDP-GlcA and decreased that of UDP-Glc (Supplementary Fig. 5i). As compared to results from the AAV-Ctrl injection, AAV-mediated UGDH overexpression significantly inhibited RIPK1 activation and apoptosis (Fig. 4b, c and Supplementary Fig. 5j), which was further confirmed by immunoblotting p-S166 RIPK1 and CC3 in primary hepatocytes isolated from CD-HFD-fed mice (Fig. 4d). Accordingly, liver damage as measured by serum ALT and AST concentrations and fibrosis as determined by Masson's trichrome and sirius red staining were reduced after AAV-UGDH administration relative to the controls (Fig. 4e, f). The concentration of liver

hydroxyproline and the expression of profibrotic genes were also markedly reduced by AAV-mediated UGDH overexpression as compared to that in AAV-Ctrl-injected group (Fig. 4g, h).

### UGDH controls RIPK1 activation through UDP-Glc to UDP-GlcA

Metabolic enzyme activity is commonly regulated by metabolites through allosteric regulation or feedback inhibition[41]. Metabolites have also been shown to regulate non-metabolic proteins[30]. Because the cellular UDP-Glc levels were increased while UDP-GlcA levels were decreased in *Ugdh*[f/f]*;Alb-Cre* livers as compared to *Ugdh*[f/f] control livers (Supplementary Fig. 6a), we hypothesized that UGDH regulates RIPK1 activation through UDP-Glc to UDP-GlcA. Since transporters for UDP-sugar are also located in the plasma membrane[30], we then screened the metabolites in this pathway by supplementing cells with UDP-Glc, UDP-GlcA, UDP-Xyl, and hyaluronic acid, respectively, to assess their effect on TNFα-induced apoptosis in *Ugdh*[f/f] and *Ugdh*[f/f]*;Alb-Cre* hepatocytes. Interestingly, we found that UDP-GlcA, which was fully restored to control levels in *Ugdh*[f/f]*;Alb-Cre* hepatocytes (Supplementary Fig. 6b), significantly suppressed TNFα-induced apoptosis in *Ugdh*[f/f]*;Alb-Cre* hepatocytes (Fig. 5a and Supplementary Fig. 6c). In contrast, UDP-Xyl or hyaluronic acid did not show any effect on TNFα-induced apoptosis in both *Ugdh*[f/f] and *Ugdh*[f/f]*;Alb-Cre* hepatocytes (Supplementary Fig. 6c). Although UDP-Glc treatment significantly increased its levels in hepatocytes (Supplementary Fig. 6d), UDP-Glc only showed very minor inhibitory effect on TNFα-induced apoptosis in *Ugdh*[f/f]*;Alb-Cre* hepatocytes (Fig. 5a and Supplementary Fig. 6c). Consistently, UDP-GlcA, but not UDP-Glc, substantially inhibited RIPK1 activation as indicated by p-S166 RIPK1 and apoptosis as indicated by CC8 and CC3 in TNFα/CHX-treated UGDH-deficient hepatocytes (Fig. 5b).

We next explored whether UDP-GlcA is sufficient to inhibit RIPK1 activation in a general manner in other models with activated RIPK1, such as TNFα-induced cell death in bone-marrow-derived macrophages (BMDMs). TNFα stimulation promotes the formation of a transient intracellular complex (complex I) at TNFR1 which coordinates an intricate set of ubiquitination and phosphorylation events to control the activation of RIPK1[35,42,43]. We first examined whether UDP-GlcA regulates the activation of RIPK1 in complex I. To this end, we supplemented BMDMs with UDP-GlcA, which significantly increased the cellular levels of UDP-GlcA (Supplementary Fig. 6e), and found that UDP-GlcA treatment in BMDMs substantially inhibited the rapid activation of RIPK1 in complex I induced by TNFα (Fig. 5c). We next investigated whether UDP-GlcA inhibits RIPK1 activation in BMDMs treated with TNFα/TAK1 inhibitor 5Z7-Oxozeaenol (5z7), which is a well-established protocol to induce RIPK1 activation and RIPK1 kinase-dependent apoptosis (RDA)[44]. We found that UDP-GlcA also showed potency in reducing RDA in BMDMs treated with TNFα/5z7 (Fig. 5d), which was further confirmed by immunoblotting p-S166 RIPK1 and CC3 (Supplementary Fig. 6f). Consistent with decreased activation of RDA, we found that UDP-GlcA-treated BMDMs showed less complex II formation than that of vehicle-treated BMDMs (Supplementary Fig. 6g). In BMDMs, when caspases are inhibited by the pan-caspase inhibitor zVAD, the kinase activity of RIPK1 is activated to interact with RIPK3 to induce the formation of a RIPK1/RIPK3 complex (known as necrosome) and the activated RIPK3 in turn mediates the phosphorylation of MLKL to promote the execution of necroptosis[45,46]. We treated BMDMs with TNFα and Smac mimetic compound SM-164 in the presence of zVAD, which is a well-established protocol to induce RIPK1 activation and necroptosis[47], and found that UDP-GlcA largely reduced TNFα/SM-164/zVAD-induced necroptosis in BMDMs in a dose-dependent manner (Fig. 5e). The inhibition of RIPK1 activation and necroptosis by UDP-GlcA was also marked by the reduced levels of p-S166 RIPK1 and the necroptotic biomarkers p-T231/S232 RIPK3, and p-S345 MLKL relative to that of vehicle treatment (Supplementary Fig. 6h). Consistent with decreased activation of necroptosis, we found that UDP-GlcA-treated BMDMs showed less necrosome formation than

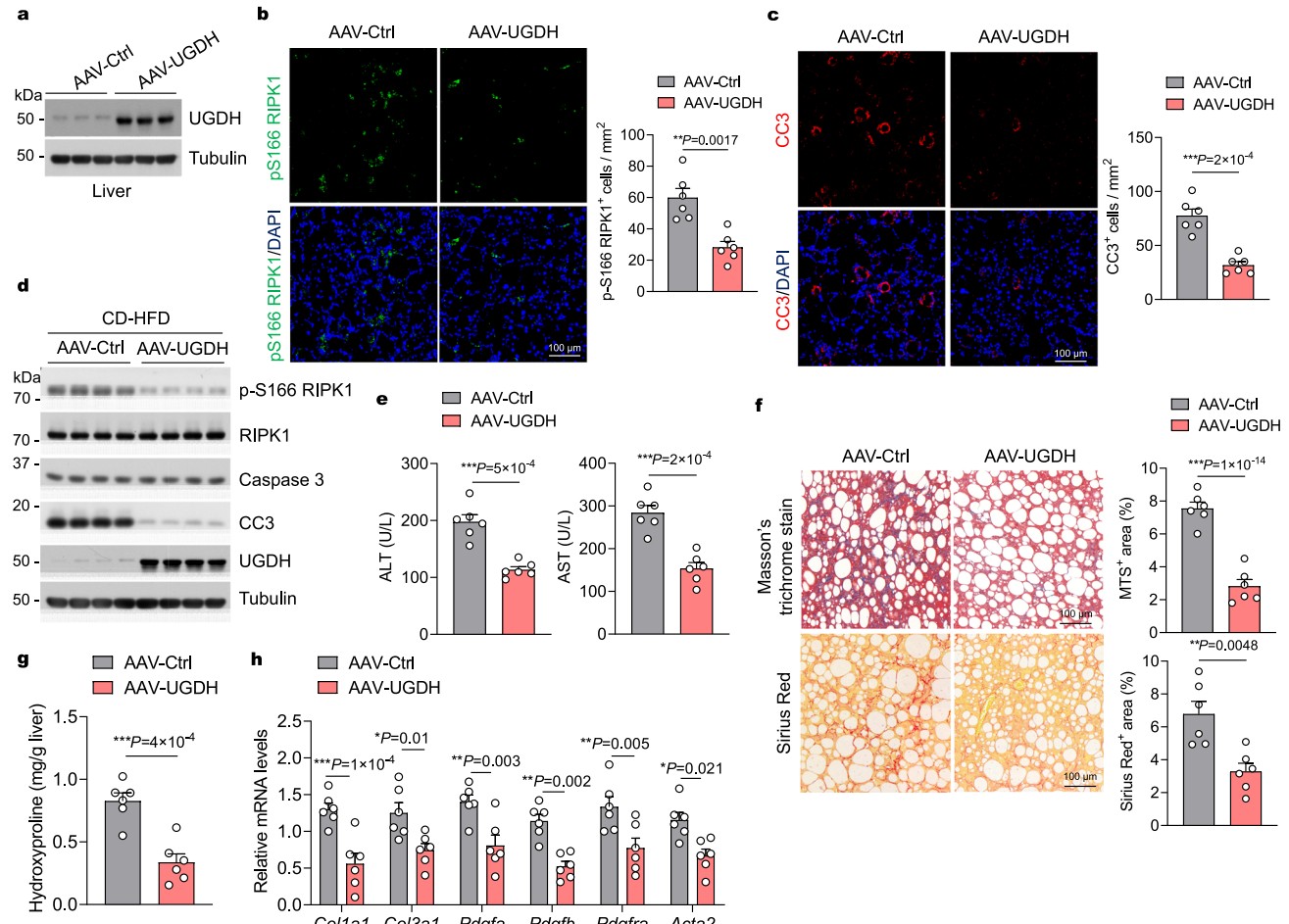

**Fig. 4 | Therapeutic effects of restoring UGDH expression on NASH in mice.**
**a** Representative immunoblotting analysis of UGDH protein levels in livers of mice that were fed CD-HFD for 4 weeks, followed by AAV-UGDH or AAV-Ctrl injection while fed continuous CD-HFD for 8 weeks. $n = 3$ mice per group. Immuno-fluorescence images of p-S166 RIPK1 (**b**) and CC3 (**c**) of liver sections from 20-week-old mice that were fed CD-HFD for 4 weeks, followed by AAV-UGDH or AAV-Ctrl injection while fed continuous CD-HFD for 8 weeks. Representative images out of $n = 6$ mice for each group are represented. Graph depicting numbers of p-S166 RIPK1$^+$ (**b**) and CC3$^+$ (**c**) cells on liver sections of indicated groups. Each dot represents an individual mouse. **d** Representative immunoblotting analysis of p-S166 RIPK1 and CC3 protein levels in primary hepatocytes isolated from livers of mice that were fed CD-HFD for 4 weeks, followed by AAV-UGDH or AAV-Ctrl injection while fed continuous CD-HFD for 8 weeks. $n = 4$ mice per group. **e** Serum

levels of ALT and AST of mice that were fed CD-HFD for 4 weeks, followed by AAV-UGDH or AAV-Ctrl injection while fed continuous CD-HFD for 8 weeks. $n = 6$ mice for each group. Each dot represents an individual mouse. **f** Representative images of Masson's trichrome and Sirius red stained liver sections of mice that were fed CD-HFD for 4 weeks, followed by AAV-UGDH or AAV-Ctrl injection while fed continuous CD-HFD for 8 weeks. Graph depicting percentage of MTS$^+$ and Sirius red$^+$ area on liver sections of indicated groups. $n = 6$ mice for each group. Each dot represents an individual mouse. Liver hydroxyproline (**g**) and mRNA expression of fibrogenic genes (**h**) in livers of mice that were fed CD-HFD for 4 weeks, followed by AAV-UGDH or AAV-Ctrl injection while fed continuous CD-HFD for 8 weeks. $n = 6$ mice for each genotype. Each dot represents an individual mouse. Mean ± s.e.m. (**b, c, e–h**). Unpaired two-tailed t-test (**b, c, e–g**). Two-way ANOVA, post hoc Bonferroni's test (**h**). Source data are provided as a Source Data file.

that of vehicle-treated BMDMs (Supplementary Fig. 6i). Although UDP-Glc treatment increased its levels in BMDMs (Supplementary Fig. 7a), UDP-Glc only showed very limited inhibitory effect on TNFα/5z7-induced RDA and TNFα/SM-164/zVAD-induced necroptosis, respectively, in BMDMs (Supplementary Fig. 7b, c). Thus, UDP-GlcA suppresses RIPK1 activation in TNFα signaling and UGDH limits RIPK1 activity by converting UDP-Glc to UDP-GlcA.

## Knockdown of UXS1 suppresses RDA in UGDH-deficient hepatocytes

To further clarify whether it is UDP-GlcA itself, and not the process of producing glycosaminoglycans through the uronic acid pathway, that inhibits TNFα-induced RIPK1 activation and apoptosis in *Ugdh*[f/f];*Alb-Cre* hepatocytes, we depleted UXS1, which catalyzes the decarboxylation of UDP-GlcA to UDP-Xyl and necessary for the biosynthesis of glycosaminoglycans[24,25], in hepatocytes. The cellular levels of UDP-GlcA were significantly increased in both *Ugdh*[f/f] and *Ugdh*[f/f];*Alb-Cre* hepatocytes after

knocking down UXS1 (Fig. 5f). Consistent with the role of UDP-GlcA in suppressing TNFα-induced RIPK1 activation and apoptosis in *Ugdh*[f/f];*Alb-Cre* hepatocytes, knockdown of UXS1 significantly reduced RIPK1 activation and apoptosis in *Ugdh*[f/f];*Alb-Cre* hepatocytes, but not in *Ugdh*[f/f] hepatocytes (Fig. 5g, h). Thus, boosting UDP-GlcA levels by reducing UXS1 expression suppresses TNFα-induced RIPK1 activation and apoptosis in *Ugdh*[f/f];*Alb-Cre* hepatocytes.

## UDP-GlcA interacts with RIPK1 kinase domain

We next investigated the possibility that UDP-GlcA directly targets and inhibits RIPK1 kinase. To this end, we performed an in vitro kinase assay to compare the ability of RIPK1 kinase domain (KD) auto-phosphorylation in the presence or absence of UDP-GlcA. We found that co-incubation of UDP-GlcA dose-dependently reduced the activation of RIPK1-KD as shown by p-S166 RIPK1 (Fig. 6a). In contrast, UDP-Glc had very minor inhibitory effect on RIPK1 kinase activation (Fig. 6a). We then examined whether UDP-GlcA interacts with RIPK1-

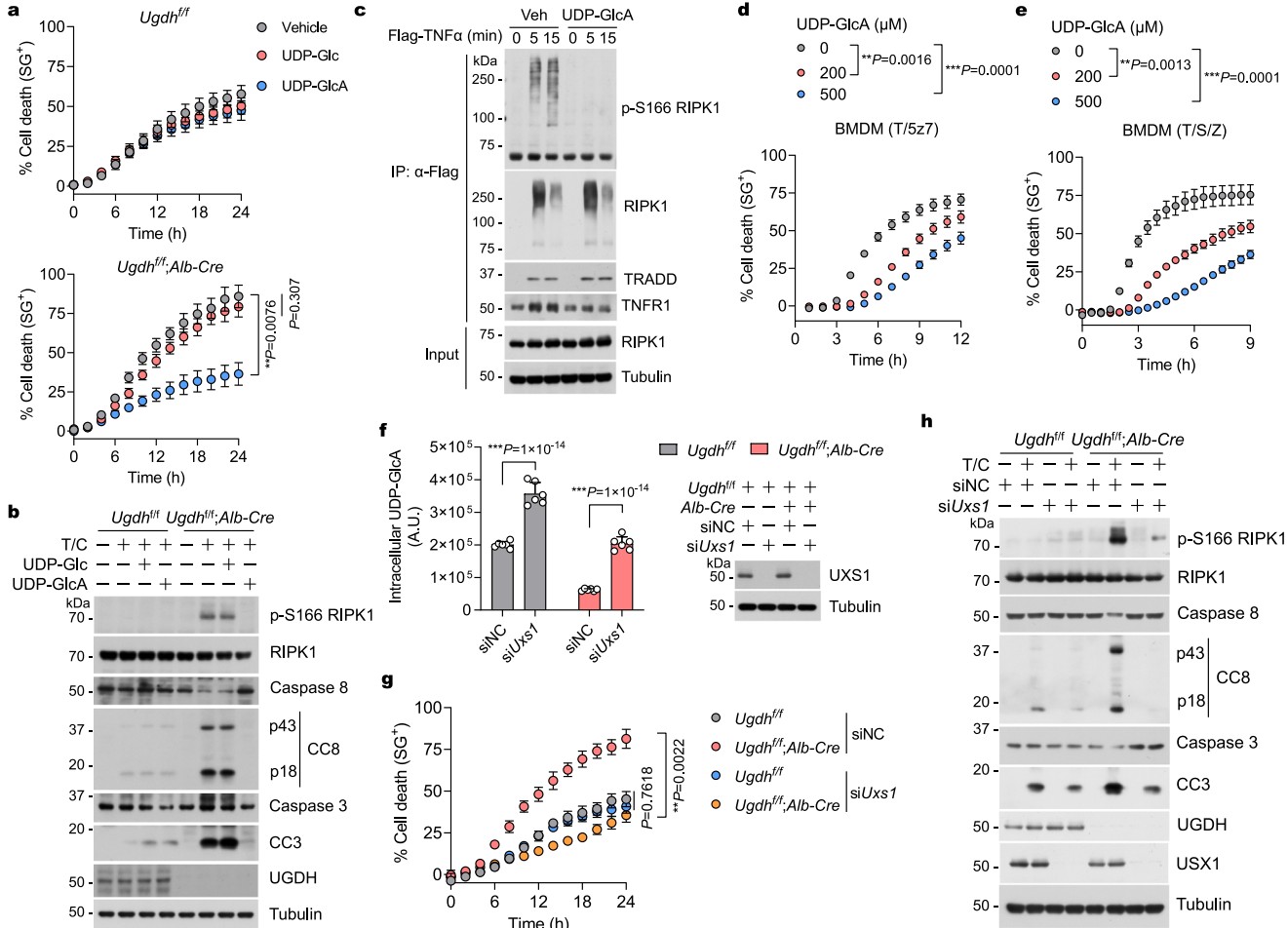

**Fig. 5 | UGDH controls RIPK1 activation through UDP-Glc to UDP-GlcA.** Primary hepatocytes were treated with CHX (1 μM)/TNFα (10 ng/ml) for indicated time (**a**) or 12 h (**b**) in the presence or absence of UDP-Glc (100 μM) or UDP-GlcA (100 μM). Cell death was measured by SytoxGreen positivity assay (**a**). The levels of p-S166 RIPK1, CC8 and CC3 were determined by immunoblotting, (**b**). **c** BMDMs were treated with Flag-TNFα (100 ng/ml) for indicated time in the presence or absence of UDP-GlcA (200 μM). TNF-RSC was immunoprecipitated using anti-Flag resin. The immune complexes were analyzed by immunoblotting using anti-p-S166 RIPK1 antibody. **d** BMDMs were treated with 5z7 (100 nM)/TNFα (1 ng/ml) for indicated time in the presence or absence of UDP-GlcA with indicated concentration. Cell death was measured by SytoxGreen positivity assay. **e** BMDMs were treated with SM-164 (S, 100 nM)/zVAD (10 μM)/TNFα (1 ng/ml) for indicated time in the

presence or absence of UDP-GlcA with indicated concentration. Cell death was measured by SytoxGreen positivity assay. **f** Primary hepatocytes were transfected with siRNAs targeting UXS1 or NC for 72 h. Intracellular UDP-GlcA concentrations were then determined. $n = 6$ mice for each group. Each dot represents an individual mouse. Knockdown efficiency is shown by immunoblotting with UXS1. **g, h** Primary hepatocytes were transfected with siRNAs targeting UXS1 or NC for 48 h. The cells were then treated with CHX (1 μM)/TNFα (10 ng/ml) for indicated time (**g**) or 12 h (**h**). Cell death was measured by SytoxGreen positivity assay (**g**). The levels of p-S166 RIPK1, CC8 and CC3 were determined by immunoblotting. $n = 3$ independent experiments (**a–e, g, h**). Mean ± s.e.m. (**a, d–g**). Two-way ANOVA (**a, d, e, g**), post hoc Bonferroni's test (**f**). Source data are provided as a Source Data file.

KD directly by using a thermal shift assay[48]. The addition of UDP-GlcA to RIPK1-KD in the thermal shift assay increased its Tm by 7.2 °C, suggesting the binding of UDP-GlcA to RIPK1-KD (Fig. 6b). Consistent with the higher potency of UDP-GlcA than UDP-Glc in suppressing RIPK1 activation, the addition of UDP-Glc to RIPK1-KD only increased its Tm by 2.7 °C (Fig. 6b).

To understand the structural basis that underlies the UDP-GlcA-regulated RIPK1 kinase activity, we carried out molecular docking to computationally generate the structure for UDP-GlcA/UDP-Glc in binding with RIPK1 kinase domain. Both UDP-GlcA and UDP-Glc bound with RIPK1-KD in similar conformations (Supplementary Fig. 7d), however, the binding affinity of UDP-GlcA with RIPK1-KD ($\Delta G = -60.3$ kCal mol$^{-1}$) was substantially higher than that of UDP-Glc with RIPK1-KD ($\Delta G = -39.5$ kCal mol$^{-1}$) (Supplementary Fig. 7d), which is consistent with our experimental results. UDP-GlcA was predicted to cover both the ATP-binding site and the allosteric site of RIPK1-KD (Fig. 6c and Supplementary Fig. 7d). UDP-GlcA forms two hydrogen bonds with D156 residue of RIPK1-KD. UDP-GlcA also forms broad

polar interactions with the solvent exposure region of RIPK1-KD, including D24, Y94, and N99 residues. Hydrophobic interactions with L157 and F162 residues of RIPK1-KD may also contribute to the binding of UDP-GlcA and RIPK1-KD (Fig. 6c).

On the basis of the results of molecular dynamics simulations, we expressed and purified several sf9-expressed recombinant RIPK1-KD mutant proteins (D156A, Y94A, F162A, and N99A/D24A). We found that mutating RIPK1-KD D156 substantially reduced its interaction with UDP-GlcA as shown by the decreased Tm values with UDP-GlcA in the thermal shift assay (Fig. 6d). Y94A mutant of RIPK1 also showed mild reduction in binding with UDP-GlcA (Fig. 6d). However, F162A single point mutation and N99A/D24A double-point mutation only slightly reduced their interaction with UDP-GlcA (Fig. 6d). These results demonstrate the importance of D156, Y94, F162, and N99/D24 in binding with UDP-GlcA. We then performed an in vitro kinase assay to compare the kinase activity of RIPK1-D156A and RIPK1 WT. D156 of the "DFG" motif is important for RIPK1's engagement of Mg$^{2+}$ ions, a necessary cofactor for ATP and enzymatic reactions[49,50]. Consistently,

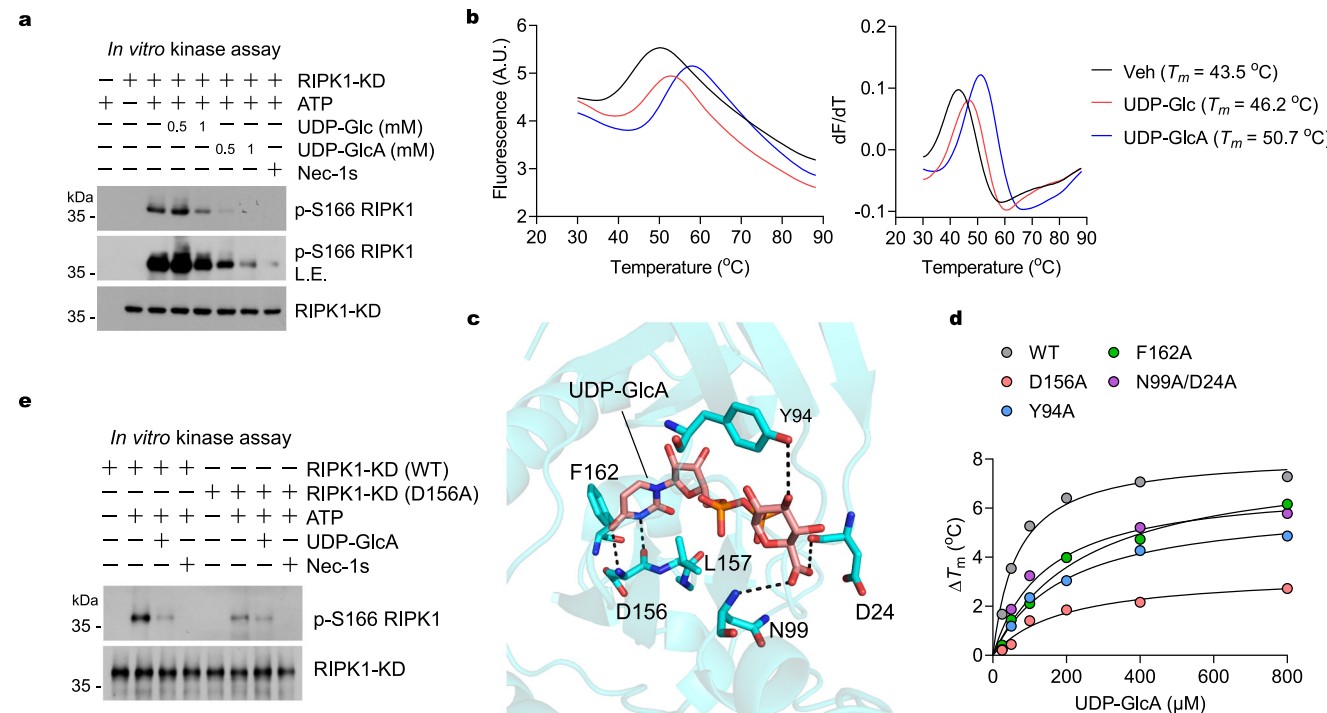

**Fig. 6 | UDP-GlcUA targets and inhibits RIPK1. a** RIPK1-KD protein (10 μM) expressed and purified from Sf9 cells were incubated with Nec-1s (50 μM) or UDP-Glc and UDP-GlcA with indicated concentrations for 1 h followed by in vitro kinase assay in the presence or absence of ATP (100 μM) at 30 °C for 0.5 h. The samples were then analyzed by immunoblotting with p-S166 RIPK1. L.E., long-time exposure. Similar results were obtained from $n = 3$ independent experiments. **b** The in vitro binding of RIPK1-KD protein (50 μM) with UDP-Glc (500 μM) and UDP-GlcA (500 μM) was determined by thermal shift assay. Data were collected in the presence of UDP-Glc or UDP-GlcA, leading to a rightward shift in the unfolding transition. The apparent melting temperature (Tm) is the peak in the derivative of the unfolding curve (dF/dT), which is used as an indicator of thermal stability. **c** The binding pose of UDP-GlcA in complex with RIPK1-KD was generated by molecular

docking. The details of the interactions between the compound and RIPK1-KD were shown. UDP-GlcA was shown as orange sticks, and the protein was shown as cartoon and colored cyan with key residues highlighted in sticks. Hydrogen bonds were shown as black dashed lines. **d** Wild-type RIPK1-KD and indicated mutant proteins (50 μM) were incubated with different concentration of UDP-GlcA. The change of Tm value was determined by thermal shift assay. **e** Wild-type RIPK1-KD and the D156A mutant protein (10 μM) were incubated with Nec-1s (50 μM) or UDP-GlcA (500 μM) for 1 h followed by in vitro kinase assay in the presence or absence of ATP (100 μM) at 30 °C for 0.5 h. The samples were then analyzed by immunoblotting with p-S166 RIPK1. Similar results were obtained from $n = 3$ independent experiments. Source data are provided as a Source Data file.

we found that D156A mutation markedly decreased the kinase activity of RIPK1-KD (Fig. 6e). In addition, UDP-GlcA was able to further inhibit RIPK1-D156A phosphorylation (Fig. 6e), suggesting that the interaction of UDP-GlcA and RIPK1-D156 mainly but not totally contribute to the interaction of UDP-GlcA and RIPK1-KD, which is consistent to the binding model that UDP-GlcA interacts with multiple sites of RIPK1-KD. *Ripk1*-knockout BMDMs complemented with D156A RIPK1 also showed reduced sensitivity to RIPK1-dependent cell death as compared to that of WT RIPK1 (Supplementary Fig. 7e, f). In addition, consistent with the role of D156A in forming a hydrogen bond with UDP-GlcA, the addition of UDP-GlcA was unable to provide additional protection in *Ripk1*-deficient cells expressing the D156A RIPK1 mutant (Supplementary Fig. 7e, f). Together, these results suggest that UDP-GlcA can bind the kinase domain of RIPK1 to inhibit its activation and prevent RIPK1-dependent cell death.

## UDP-GlcA therapeutically improve liver damage by suppressing RIPK1

After elucidating the role of UDP-GlcA in suppressing TNFα-induced RIPK1 activation and apoptosis, we next evaluated the potential therapeutic effect of UDP-GlcA against NASH-associated liver damage in mice. We fed mice with CD-HFD for 8 weeks to establish advanced NASH and then injected mice with vehicle or UDP-GlcA by tail-vein injection for 4 weeks while continuing CD-HFD (Supplementary Fig. 8a). Injection of 200 mg/kg UDP-GlcA every two days for 4 weeks in CD-HFD-fed mice significantly increased cellular levels of UDP-GlcA in the livers of CD-HFD-fed mice (Supplementary Fig. 8b). UDP-GlcA

treatment had no effect on body weight and hepatic weight (Supplementary Fig. 8c), or hepatic steatosis and hepatic TG, TC and NEFA levels (Supplementary Fig. 8d, e), or macrophage infiltration and pro-inflammatory markers expression in livers (Supplementary Fig. 8f, g), but significantly reduced cell death in CD-HFD-fed mice (Supplementary Fig. 8h). Consistent with the role of UDP-GlcA in suppressing RIPK1, UDP-GlcA injection substantially reduced RIPK1 activation and apoptosis as determined by immunoblotting of p-S166 RIPK1 and CC3, respectively, in primary hepatocytes isolated from CD-HFD-fed mice (Fig. 7a). The inhibition of RIPK1 and apoptosis by UDP-GlcA treatment was further validated by immunostaining of p-S166 RIPK1 and CC3, respectively, in liver sections from CD-HFD-fed mice (Fig. 7b, c). Relative to vehicle-injected mice, mice that injected with UDP-GlcA showed a considerable reduction in serum ALT and AST levels (Fig. 7d). Furthermore, histological fibrosis and hydroxyproline levels in the liver were significantly reduced by UDP-GlcA treatment (Fig. 7e, f). Consistently, the disturbance of the fibrotic gene profile was greatly improved by UDP-GlcA treatment (Fig. 7g). These results indicate that treatment of UDP-GlcA has the ability to prevent NASH-associated liver damage and fibrosis by suppressing RIPK1 activation and hepatocyte apoptosis, and thus UDP-GlcA may represent a promising agent for the therapy of theses pathological conditions.

## Reducing UXS1 expression improves NASH-associated liver damage

Because depletion of UXS1 increased the levels of UDP-GlcA in UGDH-deficient hepatocytes and suppressed TNFα-induced RIPK1 activation

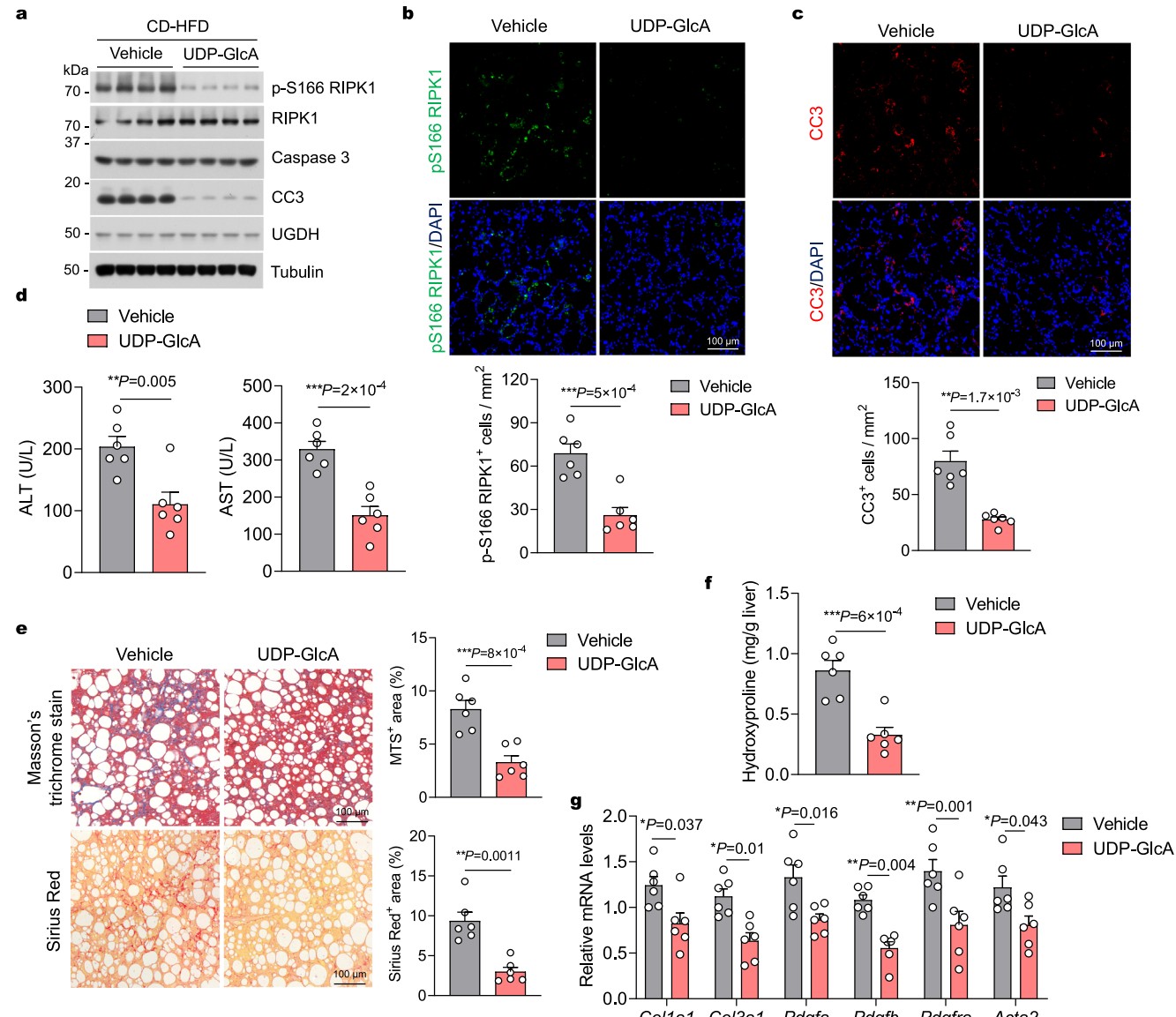

**Fig. 7 | UDP-GlcA therapeutically improve NASH-associated liver damage.**
**a** Representative immunoblotting analysis of p-S166 RIPK1 and CC3 protein levels in primary hepatocytes isolated from livers of mice that were fed CD-HFD for 8 weeks, followed by intravenous injection of 200 mg/kg UDP-GlcA or vehicle every two days for 4 weeks while fed continuous CD-HFD. $n = 4$ mice per group. Immuno-fluorescence images of p-S166 RIPK1 (**b**) and CC3 (**c**) of liver sections from 20-week-old mice that were fed CD-HFD for 8 weeks, followed by intravenous injection of 200 mg/kg UDP-GlcA or vehicle every two days for 4 weeks while fed continuous CD-HFD. Representative images out of $n = 6$ mice for each group are represented. Graph depicting numbers of p-S166 RIPK1+ (**b**) and CC3+ (**c**) cells on liver sections of indicated groups. Each dot represents an individual mouse. **d** Serum levels of ALT and AST of mice that were fed CD-HFD for 8 weeks, followed by intravenous injection of 200 mg/kg UDP-GlcA or vehicle every two days for 4 weeks while fed continuous CD-HFD. $n = 6$ mice for each group. Each dot represents an individual mouse. **e** Representative images of Masson's trichrome and Sirius red stained liver sections of mice that were fed CD-HFD for 8 weeks, followed by intravenous injection of 200 mg/kg UDP-GlcA or vehicle every two days for 4 weeks while fed continuous CD-HFD. Graph depicting percentage of MTS+ and Sirius red+ area on liver sections of indicated groups. $n = 6$ mice for each group. Each dot represents an individual mouse. **f, g** Liver hydroxyproline (**f**) and mRNA expression of fibrogenic genes (**g**) in livers of mice that were fed CD-HFD for 8 weeks, followed by intra-venous injection of 200 mg/kg UDP-GlcA or vehicle every two days for 4 weeks while fed continuous CD-HFD. $n = 6$ mice for each genotype. Each dot represents an individual mouse. Mean ± s.e.m. (**b**–**g**). Unpaired two-tailed $t$-test (**b**–**f**). Two-way ANOVA, post hoc Bonferroni's test (**g**). Source data are provided as a Source Data file.

and apoptosis, we next tested the potential of UXS1 as a therapeutic target in NASH. UXS1 levels were not altered in livers of mice after feeding CD-HFD (Supplementary Fig. 9a). We then generated AAV8 vectors encoding short hairpin RNA (shRNA) targeting *Uxs1* to induce UXS1 knockdown in livers of CD-HFD-fed mice. We fed mice with CD-HFD for 4 weeks to establish NASH and then intravenously injected mice with AAV-shRNA-*Uxs1* or scrambled control (AAV-shRNA-NC) while continuing CD-HFD for additional 8 weeks (Supplementary Fig. 9b). AAV-shRNA-*Uxs1* mice showed significant reduction of both UXS1 mRNA levels and protein levels (Fig. 8a). UXS1 knockdown also

restored the cellular levels of UDP-GlcA in livers of CD-HFD-fed mice (Fig. 8b). Knockdown of UXS1 had no effect on hepatic steatosis and hepatic TG, TC and NEFA levels (Supplementary Fig. 9c, d), or mac-rophage infiltration and pro-inflammatory markers expression in livers of CD-HFD-fed mice (Supplementary Fig. 9e, f). Consistent to its ability to regulate hepatocyte apoptosis in vitro, UXS1 knockdown sig-nificantly reduced cell death in CD-HFD-fed mice (Supplementary Fig. 9g). Knockdown of UXS1 markedly reduced RIPK1 activation and apoptosis as determined by immunoblotting of p-S166 RIPK1 and CC3, respectively, in primary hepatocytes isolated from CD-HFD-fed mice

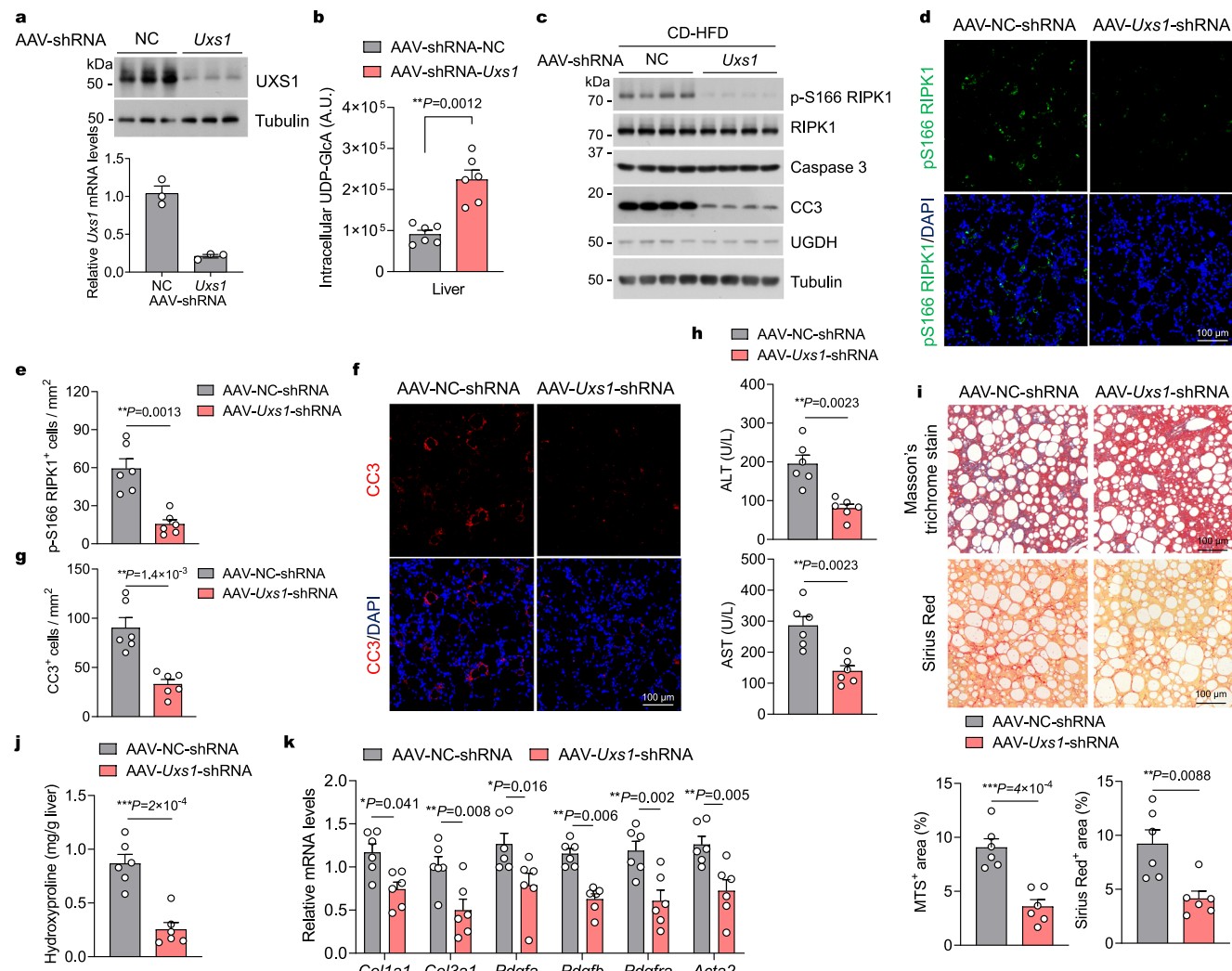

**Fig. 8 | UXS1 knockdown attenuates NASH-associated liver damage.**
**a** Representative immunoblotting analysis of UXS1 protein levels (top panel) and quantitative RT-PCR analysis of UXS1 mRNA levels (bottom panel) in livers of mice that were fed CD-HFD for 4 weeks, followed by AAV-shRNA-*Uxs1* or AAV-shRNA-NC injection while fed continuous CD-HFD for 8 weeks. $n = 3$ mice per group. **b** Mice were fed CD-HFD for 4 weeks, followed by AAV-shRNA-*Uxs1* or AAV-shRNA-NC injection while fed continuous CD-HFD for 8 weeks. Intracellular UDP-GlcA concentrations were then determined. $n = 6$ mice for each group. **c** Representative immunoblotting analysis of p-S166 RIPK1 and CC3 protein levels in primary hepatocytes isolated from livers of mice that were fed CD-HFD for 4 weeks, followed by AAV-shRNA-*Uxs1* or AAV-shRNA-NC injection while fed continuous CD-HFD for 8 weeks. $n = 4$ mice per group. **d–g** Immunofluorescence images of p-S166 RIPK1 (**d**) and CC3 (**f**) of liver sections from 20-week-old mice that were fed CD-HFD for 4 weeks, followed by AAV-shRNA-*Uxs1* or AAV-shRNA-NC injection while fed

continuous CD-HFD for 8 weeks. Representative images out of $n = 6$ mice for each group are represented. Graph depicting numbers of p-S166 RIPK1$^+$ (**e**) and CC3$^+$ (**g**) cells on liver sections of indicated groups. Serum levels of ALT and AST (**h**), and representative images of Masson's trichrome and Sirius red stained liver sections (**i**) of mice that were fed CD-HFD for 4 weeks, followed by AAV-shRNA-*Uxs1* or AAV-shRNA-NC injection while fed continuous CD-HFD for 8 weeks. Graph depicting percentage of MTS$^+$ and Sirius red$^+$ area on liver sections of indicated groups (**i**). $n = 6$ mice for each group. Liver hydroxyproline (**j**) and mRNA expression of fibrogenic genes (**k**) in livers of mice that were fed CD-HFD for 4 weeks, followed by AAV-shRNA-*Uxs1* or AAV-shRNA-NC injection while fed continuous CD-HFD for 8 weeks. $n = 6$ mice for each genotype. Mean ± s.e.m. (**a, b, e, g–k**). Unpaired two-tailed t-test (**b, e, g–j**). Two-way ANOVA, post hoc Bonferroni's test (**k**). Source data are provided as a Source Data file.

(Fig. 8c). The reduction of RIPK1 activation and apoptosis upon UXS1 knockdown was also confirmed by p-S166 RIPK1 and CC3 immunostaining in liver sections from CD-HFD-fed mice (Fig. 8d–g). Furthermore, the serum levels of ALT and AST were significantly decreased in AAV-shRNA-*Uxs1* mice as compared to that in AAV-shRNA-NC mice (Fig. 8h), indicating a decline of liver damage after depleting UXS1 in CD-HFD mice. Accordingly, histological hepatic fibrosis was also mitigated by UXS1 knockdown (Fig. 8i). In addition, the levels of hydroxyproline and the expression of profibrotic markers were also significantly inhibited by AAV-shRNA-*Uxs1* injection as compared to the group that received AAV-shRNA-NC injection (Fig. 8j, k). These results further suggest that UDP-GlcA metabolism critically controls RIPK1-driven liver damage in NASH.

## Discussion

In this study, using an unbiased approach, we identify UGDH as a critical metabolic regulator of hepatocyte apoptosis in NASH. We demonstrate that UGDH suppresses hepatocyte apoptosis by inhibiting RIPK1 activation in both in vitro model of inflammation-induced hepatocellular death and in vivo model of NASH. We further show that UGDH inhibits RIPK1 activation through converting UDP-Glc into UDP-GlcA. In addition, boosting the levels of UDP-GlcA by reducing UXS1 in UGDH-deficient hepatocytes also suppressed the activation of RIPK1 and hepatocyte apoptosis. We also provide evidence supporting a role for UGDH in hepatocytes as a robust suppressor of RIPK1-driven NASH-associated liver damage that functions to resolve hepatocellular death and fibrosis. The therapeutic potential of UGDH, UXS1, and UDP-GlcA

were also verified in mouse model of NASH. Thus, our study indicates that strategies boosting the hepatocellular levels of UDP-GlcA to suppress active RIPK1 are promising leads as NASH therapies.

RIPK1 is the major downstream target mediating the TNFα signaling. There are two distinct apoptotic pathways in TNFα signaling: a RIPK1 kinase-independent pathway (RIA) that is inhibited by NF-κB/cFLIP and a RIPK1 kinase-dependent pathway (RDA) that is inhibited by NF-κB-independent mechanisms[51]. RIPK1 kinase plays an important role in liver disease pathogenesis by regulating caspase-dependent hepatocyte apoptosis induced by TNFα. Hepatocyte-specific deletion of endogenous inhibitor of RIPK1 kinase, such as TAK1—which performs inhibitory phosphorylation on RIPK1—leads to spontaneous hepatocyte apoptosis, steatohepatitis, fibrosis and hepatocellular carcinoma (HCC)[52,53]. Inactivating RIPK1 kinase by introducing RIPK1-D138N mutation in hepatocellular TAK1-deficient mice significantly ameliorated liver pathology by preventing hepatocyte apoptosis and subsequently reduced HCC development[53]. In this study, we identify UGDH as an endogenous inhibitor of RIPK1 in hepatocytes. Loss of UGDH converts RIA into RDA due to unleased RIPK1 activity in TNFα-stimulated hepatocytes. Unlike hepatocellular TAK1-deficient mice, mice with hepatocyte-specific loss of UGDH appear normal and do not show liver pathology in unstressed condition. However, in CD-HFD-induced NASH, hepatocyte-specific loss of UGDH exaggerates NASH-associated liver damage and fibrosis by promoting hepatocyte apoptosis, which is ameliorated by RIPK1-D138N mutation. In patients with NASH, hepatocyte apoptosis is a major contributor to liver damage and fibrosis, which are the main determinants of mortality in NASH[6,10,11]. With ongoing advances of increasing number of RIPK1 inhibitors in human clinical studies for the treatment of many human inflammatory and degenerative diseases[16], we expect that preventing hepatocyte apoptosis by RIPK1 inhibitors may represent a promising approach to treat NASH-associated liver damage and fibrosis.

In addition to apoptosis, necroptosis has also been implicated in NAFLD and NASH in both animal models and human disease[54,55]. The essential initiator and effector of necroptosis are RIPK3 and MLKL, respectively. RIPK1 kinase activity can induce both apoptosis and necroptosis[15]. For liver disease pathogenesis, however, strong evidence has been obtained only for the involvement of RIPK1 kinase-dependent apoptosis, while the relevance of necroptosis remains debatable[56]. Indeed, robust RIPK3 expression is essential for necroptosis, yet several publications indicate that hepatocytes do not express RIPK3 protein[57–59]. A recent study revealed that RIPK3 is epigenetically silenced in both human and mouse hepatocytes, affecting their ability to undergo necroptosis[59]. In the setting of human NASH, *RIPK3* mRNA is repressed or at least not overtly expressed across paired human NASH biospecimens collected before and after dietary intervention. Furthermore, in HFD and CD-HFD-induced NAFLD and NASH mouse models, steatohepatitis, liver pathology, liver damage, fibrosis, and apoptosis were all similar between wild-type and MLKL-deficient mice. Consistently, RNAscope experiments could not detect expression of *Ripk3* mRNA in either WT or MLKL-deficient animals in NAFLD and NASH[59]. In this study, we found that UGDH deficiency did not induce necroptosis in hepatocytes treated with TNFα despite the activation of RIPK1. Thus, hepatocyte necroptosis may not contribute to the progression of NASH-associated liver damage.

UGDH is the key enzyme that catalyzes production of the precursor for hyaluronic acid (HA). The function of UGDH is largely characterized in human cancers. UGDH frequently overexpressed in various cancers and shown to be involved in cancer progression through both HA-dependent and independent mechanisms[30,60–62]. In present study, we demonstrate a function of UGDH as a suppressor of TNFα-induced hepatocyte apoptosis in NASH. We show that UGDH is progressively decreased in proportion to NASH severity. We further

show that UGDH inhibits TNFα-induced hepatocyte apoptosis through a HA-independent mechanism, in which UGDH indirectly inhibits RIPK1 activation by converting UDP-Glc into UDP-GlcA. UDP-GlcA has been considered to be primarily metabolic intermediate that acts as activated carrier of sugar moieties for enzymatic biosynthesis of polysaccharides containing glucuronate[63]. In this study, we reveal a molecular function of UDP-GlcA, which directly binds to RIPK1 kinase domain and prevents RIPK1 from being activated by auto-phosphorylation, and attenuating RIPK1 kinase-dependent cell death. Thus, our study builds a direct link between UDP-glucuronate metabolism and RIPK1 kinase regulation.

Considering that UGDH protein, but not mRNA, levels were markedly decreased under conditions of NASH, we surmised that there was post-transcriptional regulation of UGDH during NASH pathogenies, such as ubiquitination-dependent degradation. In line with this notion, UGDH have been found to have multiple ubiquitination sites (PhosphoSitePlus). In addition, UGDH may also interact with multiple E3 ubiquitin protein ligase (BioGrid). Future study using in vivo NASH mouse models to screen for upstream regulators of UGDH protein levels, such as E3 ubiquitin protein ligase, is needed.

Taken together, results from our study uncover that NASH-associated UGDH reduction promotes RIPK1 activation due to reduced synthesis of UDP-GlcA, which contributes to the pathogenesis of NASH-associated liver damage through RIPK1 kinase-dependent hepatocyte apoptosis. The findings establish a working model encompassing the function of UDP-GlcA metabolism in NASH and reveal promising drug targets and agents for treatment of this common disease.

## Methods

### Ethics statement
Animal experiments were conducted according to the protocols approved by the Standing Animal Care Committee at the Huazhong University of Science and Technology Tongji Medical College. All procedures involving human samples were approved by the Ethics Committee of Xinhua Hospital affiliated to Shanghai Jiao Tong University School of Medicine (Approval No. XHEC-D-2022-237), and were consistent with the principles outlined in the Declaration of Helsinki.

### Animals
*Ugdh*flox/flox mice were from Cyagen, China (S-CKO-06534). Alb-Cre mice were from The Jackson Laboratory (Catalog No. 003574). *Ripk1*D138N/D138N mice were generated as previous reported[53]. All mice were in the C57BL/6 J background. *Ugdh*flox/flox;*Alb-Cre* mice were crossed with *Ripk1*D138N/D138N mice to generate *Ugdh*flox/flox;*Alb*-Cre;*Ripk1*D138N/D138N mice. For choline-deficient high-fat-diet (CD-HFD)-induced NASH mouse model, 8-week-old male mice of indicated genotypes were fed CD-HFD (60% Fat, 0.1% Methionine and no added Choline, Research Diet, Cat. A06071302) or AMLN diet (40% Fat, 20% Fructose and 2% cholesterol, Research Diet, Cat. D09100301) for 8-12 weeks. For HFD-induced NAFLD mouse model, 8-week-old male mice of indicated genotypes were fed HFD (60% Fat, Research Diet, Cat. D12492) for 16 weeks. All animals were maintained in a specific pathogen-free environment and housed with no more than five animals per cage under controlled light (12-hour light and 12-hour dark cycle), temperature ($24 \pm 2\,°C$) and humidity ($50 \pm 10\%$) conditions, and provided with *ad libitum* access to food and water throughout all experiments. Local and federal regulations regarding animal welfare were followed. General welfare monitoring was performed on a daily basis. Prior to individual experiments, each animal was diligently checked for its suitability according to preset criteria approved by the local animal welfare authorities. Prior to liver harvest, mice were euthanized by isoflurane overdose followed by cervical dislocation. This is an approved method according to the recommendations of the panel on Euthanasia of the American Veterinary Medical Association.

## Human liver samples

Steatotic livers were obtained from individuals (age $47.1 \pm 12.5$ years) with NAFLD or NASH who underwent liver biopsy or steatotic liver transplantation. Liver steatosis due to excessive alcohol consumption (>140 g per week for men or >70 g for women), use of toxins or drugs, and viral infections (eg, hepatitis B virus and hepatitis C virus) were excluded from the study. Nonsteatotic liver samples were collected from the normal donor livers (age $48.7 \pm 12.4$ years). All donor livers were allocated through China Organ Transplant Response System from 2017 to 2021. Donors were enrolled in the study voluntarily, and the families of the organ donors were asked for consent. Written informed consent was obtained from the subjects or families of all participants. Hierarchical steatosis and steatohepatitis were independently diagnosed by two pathologists according to the Standard Histological Criteria Rating System established by the NASH Clinical Research Network. Cases with NASH activity scores (NAS) of 1–3, and ballooning scores of 0 and no fibrosis were classified as simple steatosis. Cases with NAS > 4 or NAS 3-4 but with fibrosis were classified as NASH. Cases with a NAS of 0 were classified as normal.

## Cell culture

Primary hepatocytes were cultured in RPMI-1640 (Gibco) with 10% (vol/vol) FBS (Gibco) and 100 units/ml penicillin/streptomycin (Invitrogen). BMDMs and HEK293T cells were cultured in DMEM (Gibco) with 10% (vol/vol) FBS and 100 units/ml penicillin/streptomycin. All cells were cultured in 37 °C with 5% $CO_2$.

## Primary hepatocyte isolation

6-week-old male mice were used for isolation of primary hepatocytes unless indicated. Mice were anesthetized with 90 mg/kg body weight of pentobarbital sodium first. Livers were fully digested via portal vein perfusion using Liver Perfusion Medium (17701-038; Life Technologies) and then Liver Digest Medium (17703-034; Life Technologies) at a rate of 2 ml/min for 5 min for each medium. After digestion, liver was excised, minced and filtered through a 70 μm cell strainer (352350; Falcon). Primary hepatocytes were then separated via centrifugation at $50 \times g$ for 5 min and purified on 50% Percoll solution (P1644; Sigma).

## Generation of knockout and reconstitution lines

RIPK1-knockout BMDMs were generated using CRISPR/Cas9 system-mediated gene knockout method with lentivirus transduction of guide RNA against mouse *Ripk1* (5′-CAC CGA CCT AGA CAG CGG AGG CTT C-3′) in the Lenti-CRISPR v2 lentiviral background and selected with 4 μg/ml puromycin. The D156A mutant of RIPK1 was created by PCR-based site-directed mutagenesis, and then cloned into the pMSCV retroviral vector. RIPK1 WT or D156A mutant BMDMs were generated by stably reconstituting *Ripk1*-KO BMDMs with retrovirus transduction of RIPK1 WT or D156A mutant constructs (with PAM mutated to avoid additional editing by Cas9) and selected with 200 μg/ml hygromycin.

## Lentivirus and retrovirus production and infection

For lentiviral production, HEK293T cells were transfected with Lenti-CRISPR-v2 vector carrying sgRNA targeting RIPK1 with pMD2.G and psPAX plasmids for 48 h. For retroviral production, HEK293T cells were transfected with pMSCV vector carrying cDNA encoding RIPK1 (WT and D156A) with VSVG and GAG plasmids for 48 h. Harvested supernatant media from transfected HEK293T cells was filtered through a 0.45 μm filter. Filtered media containing lentivirus or retrovirus particles was used to infect target cells in the presence of polybrene (8 mg/ml). After 6 h of incubation with the virus, the medium was replaced with fresh medium and selected with puromycin or hygromycin after 24 h of infection.

## Adeno-associated virus 8 construction and injection

The AAV8 delivery system was used to overexpress UGDH or knockdown UXS1 in the livers of mice. AAV expressing UGDH (pAAV8-CBh-UGDH-3×FLAG-tWPA) or empty control (pAAV8-CBh-3×FLAG-tWPA) and expressing shRNA specific to mouse *Uxs1* (pAAV8-U6-shRNA-*Uxs1*-WPRE-SV40pA) or scrambled nontargeting negative control (pAAV8-U6-shRNA-NC-WPRE-SV40pA) were generated by transfecting three plasmids (pAAV flanked by the AAV inverted terminal repeat sequences, pAAV8 trans-plasmid with the AAV rep and cap genes, and the pAAV helper plasmid) in HEK293T cells. The AAV titers were $1 \times 10^{12}$ V.G./ml. Mice were injected via the tail vein with 100 μl of virus containing $2 \times 10^{11}$ V.G. of the AAV8 vector genomes. All AAVs were purchased from Taitool Biotech (Shanghai).

## Immunoblotting

Antibodies against the following proteins were used for western blot analysis: p-S166 RIPK1 (Biolynx, BX60008, 1:1000), RIPK1 (CST, 3493, 1:1000), TNFR1 (CST, 13377, 1:1000), cleaved Caspase-3 (CST, 9661, 1:1000), cleaved Caspase-8 (CST, 8592, 1:1000), RIPK3 (CST, 95702, 1:1000, 1:500 for IP), p-T231/S232 RIPK3 (CST, 91702, 1:1000), MLKL (CST, 37705, 1:1000) p-S345 MLKL (Abcam, ab196436, 1:1000), FADD (Abcam, ab124812, 1:1000), FADD (Santa Cruz, SC-6036, 1:1000 for IP), Caspase-3 (Proteintech, 19677-1-AP, 1:1000), Caspase-8 (Proteintech, 13423-1-AP, 1:1000), UGDH (Proteintech, 13151-1-AP, 1:1000), UXS1 (Invitrogen, PA5-31629, 1:1000), β-Tubulin (TRANS, HC101-02, 1:10,000). The signals were detected by Immobilon ECL Ultra Western HRP Substrate (Millipore). The membranes were reprobed after incubation in Restore Western Blot stripping buffer (21063, Thermo).

## siRNA transfection

For siRNA transfection, primary hepatocytes were transfected with 50 nM siRNA using Lipofectamin RNAiMax (Invitrogen) for 48 h following the manufacturer's instruction. The sense sequences of siRNAs used in this study were as follows: si*Ugdh*-1 (5′-GCAGUGUCAUUGCUCACAUU-3′); si*Ugdh*-2 (5′-GCAGCAGACCUGAAGUAUAUU-3′); si*Uxs1* (5′-GCCUUGUCUCUUCAGUCAAUU-3′). The knockdown efficiency was examined using western blotting.

## Analysis of cytotoxicity and viability

For rates of cell death, primary hepatocytes and BMDMs were seeded day before at 2500 cells per well in a 384-well plate. The next day, cells were pre-treated with the indicated compounds for 1 h and then stimulated with TNFα in the presence of 5 mM SytoxGreen (Invitrogen). SytoxGreen intensity were measured at intervals of 30 min using a SYNERGY H1 microplate reader (BioTek), with an excitation filter of 485 nm, emission filter of 520 nm. Data was collected by using Gen5 software (version 3.08.01, BioTek). Percentage of cell death was calculated as (induced fluorescence − background fluorescence)/(max fluorescence − background fluorescence) × 100. The maximal fluorescence is obtained by full permeabilization of the cells using Triton X-100 at a final concentration of 0.1%.

## siRNA screen

siRNA screen was performed using 114 siRNA pools (GenePharma) targeting 114 metabolic enzymes. Primary hepatocytes cultured in 384-well white plates were transfected with 50 nM siRNA by reverse transfection method using RNAiMax (Invitrogen) according to a protocol from the manufacturer. At 48 h after the transfection, the cells were treated with 1 μM cycloheximide (CHX) and TNFα (10 ng/ml) and cultured for an additional 24 h to induce apoptosis. Viability was measured using luminescence-based ATP levels as a surrogate marker in surviving cells using CellTiterGlo ATP assay (Promega). Positive control (Caspase-8) and negative control (nontargeting control siRNA,

GenePharma) were present in the same plate. The screen was performed in triplicate. Z-scores were calculated based on plate median (negative controls excluded) and median absolute deviation, with Z-score = (cell ATP value − median plate ATP value)/(plate median absolute deviation × 1.4826)[64]. The screen hits were selected based on the median Z-score of the triplicate plates with cut-offs set at Z-score > 2.576 or <−2.576.

## Immunoprecipitation

For TNF-RSC (complex I) immunoprecipitations, BMDMs were seeded in 15 cm dishes and treated as indicated with Flag-TNFα (100 ng/ml). To terminate treatment, media was removed and plates were washed two times in ice-cold PBS. Cells were lysed in 1 ml 0.5% NP-40 lysis buffer (50 mM Tris-HCl pH 7.5, 150 mM NaCl and 0.5% NP-40, 5% glycerol) supplemented with phosphatase and protease inhibitors and N-Ethylmaleimide (2.5 mg/ml, Sigma). Cell lysates were rotated at 4 °C for 30 min then clarified at 4 °C at 12,000 × $g$ for 15 min. Proteins were immunoprecipitated from cleared protein lysates with 20 μl of anti-Flag M2 beads (Sigma) with rotation overnight at 4 °C. 3× washes in 0.5% NP-40 buffer were performed, and samples were eluted by boiling in 50 μl 1× SDS loading buffer and analyzed by western blotting.

For complex II immunoprecipitations, hepatocytes and BMDMs were seeded in 10 cm dishes and treated as indicated. Cells were washed two times in ice-cold PBS before lysis in 1 ml 0.5% NP-40 lysis buffer. Cell lysates were rotated at 4 °C for 30 min then clarified at 4 °C at 12,000 × $g$ for 15 min, and the supernatant was then incubated overnight at 4 °C with 2 μl anti-FADD (Santa Cruz, SC-6036, for complex II) or mouse anti-RIPK3 antibodies (CST, 95702, for necrosome). Then the immunocomplex was captured by protein A/G agarose (Invitrogen) for 4 h at 4 °C. The immune complexes were then eluted in 1 × SDS loading buffer and analyzed by western blotting.

## Quantitative reverse-transcription PCR

Total RNA was isolated from livers of adult mice using FastPure Cell/Tissue Total RNA Isolation Kit (Vazyme). RNA concentration was measured using the Nanodrop spectrophotometer (Thermo scientific). cDNA was prepared with HiScript III RT SuperMix kit (Vazyme). cDNA (10 ng) of each sample was used for quantitative PCR with ChamQ Universal SYBR qPCR Master Mix (Vazyme). Quantitative reverse-transcription PCR of indicated genes were performed with QuantStudio 6 Flex Real-Time PCR System (Applied Biosystems). Data was collected by using QuantStudio 12K Flex software version 1.3 (Applied Biosystems). Data were analyzed according to the ΔCT method. The sequences of gene-specific primers used for PCR are shown below.

Mouse *Tnf* forward: CCCTCACACTCAGATCATCTTCT, reverse: GCTACGACGTGGGCTACAG; *Il1b* forward: GCAACTGTTCCTGAAC TCAACT, reverse: ATCTTTTGGGGTCCGTCAACT; *Ccl2* forward: TT AAAAACCTGGATCGGAACCAA, reverse: GCATTAGCTTCAGATTTA CGGGT; *Col3a1* forward: CTGTAACATGGAAACTGGGGAAA, reverse: CCATAGCTGAACTGAAAACCAC; *Acta2* forward: GTCCCAGACAT CAGGGAGTAA, reverse: TCGGATACTTCAGCGTCAGGA; *Tgfb* forward: CTCCCGTGGCTTCTAGTGC, reverse: GCCTTAGTTTGGACAGGAT CTG; *Pdgfa* forward: GAGGAAGCCGAGATACCCC, reverse: TGCTG TGGATCTGACTTCGAG; *Pdgfra* forward: TCCATGCTAGACTCAG AAGTC, reverse: TCCCGGTGGACACAATTTTTC; *Pdgfb* forward: CA TCCGCTCCTTTGATGATCTT, reverse: GTGCTCGGGTCATGTTC AAGT; *Col1a1* forward: GCTCCTCTTAGGGGCCACT, reverse: CCACGT CTCACCATTGGGG; *Alb* forward: TGCTTTTTCCAGGGGTGTGTT, reverse: TTACTTCCTGCACTAATTTGGCA; *Clec4f* forward: GAGGC CGAGCTGAACAGAG, reverse: TGTGAAGCCACCACAAAAAGAG; *Tek* forward: GAGTCAGCTTGCTCCTTTATGG, reverse: AGACACAAG AGGTAGGGAATTGA.

## Protein expression and purification

Wild type or mutants of the RIPK1 kinase domain (residues 1-312) constructs were subcloned into the Nde I and Xho I sites of pFastBac (Invitrogen) with an N-terminal 6 × His tag and an engineered cleavage site for the caspase drICE. The cleavage site of drICE is Asp-Glu-Val-Asp-Ala, and cleavage occurs between Asp and Ala. The linker between the drICE cleavage site and RIPK1 is Gly-Ser-Gly. Bacmids were generated in DH10Bac cells, and the resulting baculoviruses were generated and amplified in Sf9 insect cells. After infection by baculoviruses for 48 h, the cells were harvested in a buffer containing 25 mM Tris (pH 8.0) and 150 mM NaCl. The RIPK1 kinase domain was purified to homogeneity by nickel affinity chromatography (QIAGEN), anion-exchange chromatography (Source-15Q, GE Healthcare), and gel-filtration chromatography (Superdex-200, GE Healthcare). An additional step of drICE cleavage was performed to remove the 6 × His tag just prior to gel filtration. The purified RIPK1 was in a buffer containing 25 mM Tris (pH 8.0), 150 mM NaCl, and 5 mM DTT.

## In Vitro kinase activity assay

For the in vitro kinase activity assay, wild-type or D156A mutant of RIPK1-KD were used. RIPK1-KD protein at 1 μM was incubated in 50 μl reaction buffer containing 25 mM HEPES (pH 7.0), 10 mM MgCl₂, 50 mM NaCl, and 1 mM DTT for 30 min at 30 °C in the presence of varying concentrations of indicated compounds. For these assays, compounds were diluted to appropriate concentrations in DMSO and added in the assay systems with a final concentration of 1% DMSO. Kinase reactions were initiated by addition of 100 μM ATP, and the reactions were carried out at 30 °C for 30 min. Reactions were stopped by adding SDS-PAGE sample buffer and subjected to SDS-PAGE. RIPK1 activity was determined by immunoblotting p-S166 RIPK1.

## Thermal shift assay

To determine stability, RIPK1-KD were made to a final concentration of 1 μg/μl. SYPRO Orange dye was added to the protein to make a final concentration of 2×. UDP-GlcA and UDP-Glc were added in the mix with a final concentration as indicated and incubated at 4 °C for 1 h. The experiments were performed in 384-well plates specific for real-time PCR instrument with a total volume of 20 μl/well. The assay plate was covered with a sheet of optically clear adhesive to seal each well. The assay plate was placed into the Applied Biosystems QuantStudio 6 Real-Time PCR System. The reaction was run from 25 °C, ramping up in increments of 0.05 °C/s to a final temperature of 95 °C with fluorescence detection throughout the experiment to generate a dataset. Data was collected by using QuantStudio 12K Flex software version 1.3 (Applied Biosystems). Melting temperature of the protein (Tm) was determined by performing non-linear fitting of the dataset to a Boltzmann Sigmoidal curve in GraphPad Prism with the following equation: $Y = Bottom + (Top − Bottom)/[1 + \exp(Tm − X/Slope)]$, where $Y$ = fluorescence emission in arbitrary units; $X$ = temperature; Bottom = baseline fluorescence at low temperature; Top = maximal fluorescence at top of the dataset; Slope = describes the steepness of the curve, with larger values denoting shallower curves.

## Measurement of UDP-Glc and UDP-GlcA concentrations

Methods for the extraction of intracellular nucleotide sugars and measurements of UDP-Glc and UDP-GlcUA were described previously[30]. In brief, $3 × 10^6$ cells in each sample were collected and centrifuged at 1,000 × g for 5 min. The pellet was washed with phosphate-buffered saline (PBS). Subsequently, 200 μl ice-cold perchloric acid (0.5 M) was added to the pellet and mixed vigorously. After incubation on ice for 2 min, the samples were centrifuged at 10,000 × g at 4 °C for 5 min and the cell debris was removed. The supernatants were neutralized with 50 μl of 2.5 M KOH in 1.5 M K₂HPO₄ and incubated on ice for 2 min. The neutralized samples were centrifuged again to remove potassium perchlorate precipitate, and then filtered with

0.2 μm filters (Millipore). The cell extracts were separated by HPLC (Thermo Fisher Scientific) using a 3.9 mm × 150 mm (60 Å, 0.5 m) Nova-Pack C18 column (Thermo Fisher Scientific) with an ion-pairing mobile phase (1 ml/min) consisting of 8 mM tetrabutylammonium hydrogen sulfate, 60 mM KH$_2$PO$_4$, pH 5.3 and 15% methanol. UDP-Glc (Sigma-Aldrich) and UDP-GlcUA (Sigma-Aldrich) with purity over 99% were used as an internal standard for ion-pair HPLC. The mass spectrometry analyses were performed as follows. To evaluate the levels of UDP-Glc ($m/z = 566.30$) and UDP-GlcUA ($m/z = 610.27$) in cells or tissues, sequential quantities of the corresponding internal standard were used to generate the standard curve line. Then, the areas of ion peaks from experimental samples of UDP-Glc or UDP-GlcUA were calculated and compared to the corresponding internal standard.

## Molecular modeling and docking methods

The binding pocket was defined based on the crystal structure of RIPK1 kinase domain complexed with a high-affinity inhibitor (PDB code 7FD0). The RIPK1 part of this 3D structure served as the protein receptor in the following docking procedure performed with the molecular simulation software suite Schrödinger (version 2018-1, Schrödinger, LLC, New York, NY, 2018). The receptor was first prepared with the Protein Preparation Wizard. The structure was preprocessed following default settings except no waters were deleted at this step, then hydrogen bond assignment and restrained minimization were performed in the refinement step, followed by removing the water molecules with less than three H-bonds to non-waters. The 3D conformations of the small molecules were downloaded from PubChem (CID 8629 for UDP-Glc and CID 17473 for UDP-GlcA). The prepared structures of RIPK1 receptor and small molecules were then submitted for Glide docking to predict the binding modes. We searched potential binding poses of our small molecules at the binding site of the ligand from the prepared complex, and within a similar size of it. The standard precision mode and enhanced sampling were used for docking. To better estimate the binding affinities of our small molecules, we performed MM-GBSA using Prime in Schrödinger with the input of the docked complexes. The residues within 9 Å around the ligand were optimized. Finally, the MM-GBSA $\Delta G_{bind}$ values and their components were taken to analyze the binding affinities.

## TUNEL tissue staining

Collected mouse tissue was fixed in 4% PFA and processed for paraffin embedding. TUNEL assay was used to detect dead cells with DNA fragmentation using In Situ Cell Death Detection Kit, POD (Roche) by following manufacture's protocol.

## Biochemical serum analysis

Alanine aminotransferase (ALT) and aspartate transaminase (AST) were measured in blood serum using Cobas C111 biochemical analyzer (Roche, Mannheim, Germany) according to the manufacturer's instructions. TG, TC and NEFA contents in liver were measured using commercial kits (290-63701 for TG, 294-65801 for TC and 294-63601 for NEFA; Wako, Tokyo, Japan). Tissue TG, TC and NEFA levels were normalized to tissue weight.

## Hydroxyproline measurement

Liver tissue was homogenized in distilled water. 100 μl lysate was hydrolyzed with 100 μl 10 N NaOH at 120 °C for 1 h. Lysate was cooled on ice and neutralized with 1 N HCl. Hydroxyproline in supernatant was measured with Hydroxyproline assay kit (Abcam, Cat. Ab222941), according to the manufacturer's instruction.

## Histology and immunostaining

Tissues were harvested from mice with different genotypes and fixed with 4% paraformaldehyde. Fixed tissues were embedded with paraffin. Sections were dewaxed and then antigen retrieval was performed with 0.01 M sodium citrate. The sections were firstly blocked with 3% H$_2$O$_2$ and next with 5% goat serum and incubated with primary antibodies at 4 °C overnight and washed in PBST before incubating with secondary antibodies at RT for 2 h. To calculate the NAFLD activity score (NAS), H&E images (20×) were analyzed by a trained hepatopathologist who was blinded to the identity of the samples according to criteria described by Brunt et al.[65]. NAS is the sum of the scores of three components: steatosis (0-3), lobular inflammation (0-3) and hepatocyte ballooning (0–2). For immunostaining, fluorescent images were collected by the Leica TSC SP8 confocal microscopy system using a 20× or 40× objective. The following antibodies were used: p-S166 RIPK1 (Biolynx, BX60008, 1:500), CD45 (Servicebio, GB11066, 1:500), CC3 (CST, 9661, 1:400). Secondary antibodies: Alexa Fluor 488 goat anti-rabbit IgG (Invitrogen, A11034, 1:2000); Alexa Fluor 568 goat anti-rabbit IgG (Invitrogen, A11011, 1:2000).

## Oil red O staining

Oil red O (ORO) staining was performed to detect neutral lipids and lipid droplet morphology with slight modifications. Tissues were harvested from mice with different genotypes and fixed with 4% paraformaldehyde. The frozen tissue sections were immersed in freshly prepared 1% oil red O working solution for 10 min, counterstained with hematoxylin, and then rinsed under running tap water for 30 min. Photomicrographs were captured under a microscope.

## Fibrosis staining

Masson's trichrome stain and sirius red stain were used to determine the levels of collagen deposition. Tissues were harvested from mice with different genotypes and fixed with 4% paraformaldehyde. Fixed tissues were embedded with paraffin, sliced into 5-μm sections, and stained with Masson's trichrome or sirius red at room temperature for 2 h. All hepatic sections were examined and images were captured via light microscopy.

## Quantification and statistical analysis

All cell death data are presented as mean ± SEM of three biologically independently experiments. Mouse data are presented as mean ± SEM of the indicated $n$ values. All immunoblots were repeated at least 3 times independently with similar results. Statistical analyses were performed with GraphPad Prism 8.0 (v8.4.1). Normality of the samples were checked by using the Shapiro-Wilk test before statistical analysis. For normal distribution, either unpaired two-tailed Student's $t$-test for comparison between two groups, or one-way ANOVA with post hoc Dunnett's tests for comparisons among multiple groups with a single control was applied, while for non-normal distribution, a nonparametric statistical analysis was performed using Mann-Whitney test for two groups, or Kruskal-Wallis test followed by Dunnett's test for multiple comparisons. For cell death kinetics comparison, two-way ANOVA was applied. Multiple liner regression analysis was performed to assay the correlation of hepatic UGDH expression level with variations in NAFLD and NASH. Differences were considered statistically significant if $P < 0.05(*)$; $P < 0.01$ (**); $P < 0.001(***)$.

## Reporting summary

Further information on research design is available in the Nature Portfolio Reporting Summary linked to this article.

# Data availability

The crystal structure of RIPK1 kinase domain complexed with a high-affinity inhibitor was obtained from PDB (https://www.rcsb.org) with PDB code 7FD0. The clinical and histological characteristics of the human samples are provided with this paper. All other data generated or analyzed during this study are included in this published article and its supplementary information files. Source data are provided with this paper.

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

## Acknowledgements

The work of J.G. was supported in part by grants from the National Natural Science Foundation of China (82130020 and 82072645). The work of D.X. was supported in part by grants from the Strategic Priority Research Program of the Chinese Academy of Sciences (XDB39030600), the National Key R&D Program of China (2022ZD0213200), the National Natural Science Foundation of China (32070737 and 92049303), the Shanghai Science and Technology Development Funds (20JC1411600, 20QA1411500 and 22JC1410400), the Shanghai Key Laboratory of Aging Studies (19DZ2260400) and Shanghai Municipal Science and Technology Major Project (2019SHZDZX02).

## Author contributions

This project was conceived, designed, and directed by D.X. and J.G., T.Z. and N.Z. designed and conducted majority of the experiments. J.X. performed molecular docking experiments. S.Z. assisted with animal experiments. Y.C. assisted with in vitro experiments. The manuscript was written by D.X. and J.G.

## Competing interests

The authors declare no competing interests.
