## [Peer Review File · Nature Communications]

UDP-glucuronate metabolism controls RIPK1-driven liver damage in nonalcoholic steatohepatitisREVIEWER COMMENTS

Reviewer #1 (Remarks to the Author):

In this manuscript, the authors conducted a targeted siRNA screen using 114 siRNA pools for genes associated metabolic enzymes and identified UGDH as a suppressor of hepatocyte apoptosis induced by TNF α . In addition, they confirmed that UDP-GlcA catalyzed by UGDH directly binds to the kinase domain of RIPK1 to inhibit its activation and improve liver damage. This research reveals the novel role of UGDH and UDP-GlcA in the pathogenesis of NASH. These results are interesting and clinically meaningful. While there were some additional questions the authors might consider addressing in this paper.

1. In fig.1, the results about hepatocyte apoptosis with Nec-1 treatment is confused. The authors claimed that UGDH deficiency promotes the activation of RIPK1 and converts RIA into RDA. Whether RIA or RDA is an apoptosis form induced by TNF α , why UGDH deficiency alone has no effect on apoptosis but has a promotion on apoptosis induced by TNF α ?

If TNF α -induced apoptosis is RDA, the apoptosis of WT group with NEC-1S should be inhibited rather than unchanged as shown in the figure; If TNF α -induced apoptosis is RIA, so the apoptosis of the UGDH knockout group with NEC-1S should be returned to the same level as that of the control group rather than far lower than that shown in the figure.

2. Line 130-132, the results description should be corrected, as the data suggest necroptosis involved in the UGDH deficiency-induced cell death.

In fig.2A, the number of samples for UGDH quantitative analysis is inconsistent with that of western data.

3. In fig.3 about Ugdhf/f;Alb-Cre; Ripk1D138N/D138N, overexpression Ripk1D138N/D138N couldn't make phosphorylated RIPK1 in parent hepatocytes loss, which should be taken into consideration.

4. In fig. 3b, compared with the lane 3, the expression of PRK1 is significantly reduced. Does ugdh loss has an effect on PRK1 expression? If so, what is the reason?

5. In fig.5 & fig. 6, replenishment with UDP-GlcUA is difficult to understand. UDP-GlcUA transporters are located in the ER and not in outer cell membranes therefore there is no mechanism for its uptake into cells. How is the UDP-GA proposed to get into the cells when added to the media?

Reviewer #2 (Remarks to the Author):

In this work, the authors reported the role of UDP-glucose 6-dehydrogenase (UGDH) in NASH progression using a plethora of in vivo and in vitro rodent models. UGDH is identified as a key endogenous suppressor of RIPK1. UGDH directly interacts with RIPK3, and therefore blocks RIPK3 hyperactivation via changing UDP-glucose to UDP-glucuronate, which facilitates the UDP-glucuronate-associated inactivation of RIPK3 and balance hepatocytes metabolism homeostasis, further prevents development of CD-HFD-induced NASH phenotype.

Overall, massive amounts of data from in vivo and in vitro models of NASH have been provided to strongly support the important role of UGDH in the regulation of RIPK3 signaling and the pathogenesis of NASH. The concept of disturbance of liver metabolic homeostasis by UGDH-RIPK3 axis pathway is highly innovative. The manuscript is generally well-written, work flow is logical and quantity is impressive. The methodology employed appears to be appropriate and the experiments correctly controlled.

However, many weak points in the current work need to be clarified and performed.

My specific comments about this manuscript are as follows:

1. In first part of result, the authors determine UGDH as a key suppressor of hepatocytes apoptosis in response to TNF- α administration. TNF- α -associated two apoptotic modalities, especially RIPK1, are accordingly highlighted. A number of previous studies have confirmed that necrosome RIPK1-RIPK3 core (DOI:<https://doi.org/10.1016/j.jhep.2019.11.008>; DOI:<https://doi.org/10.1016/j.cell.2018.03.032>) play an important role in NAFLD/NASH development. However, authors do not indicate how they determined that RIPK1 is the only substrate or key substrate for UGDH. Since RIPK1-RIPK3 may synergistically participate in cell apoptosis process, the role of UGDH in RIPK3 should be performed.

The reviewer suggest the authors use mass spectrometry or multi-omics assay to determine (1) substrate specificity of UGDH-RIPK1 and (2) whether UGDH also functions on RIPK3.

2. In addition, Molecular modeling and docking method for UGDH and RIPK1 interaction assay is good one used in this work, this reviewer also prefer to observe UGDH-RIPK1 binding via protein GST pull-down assay, this detection is probably more intuitive and it also corroborates the molecular docking results.

3. In vitro experiments, primary hepatocytes and BMDMs are used to study apoptosis process, the topic of this work is to investigate role of UGDH in NASH progression by regulating RIPK1-associated hepatocyte apoptosis. Thus, liver-derived cell types may be a better choice for this study. The reviewer wonder know why the author used BMDMs instead of other liver-related cells (e.g., human HepG2 or rodent AML12). MEFs cells are used in authors' previous work (Nature Communications (2022) 13:7153, DOI:<https://doi.org/10.1038/s41467-022-34993-0>) to explore role of RIPK1 in NASH progression, but in this paper, BMDMs are used to establish genetic cells in vitro experiments. Are there any different reasons or phenomena between MEFs and BMDMs? This point need to be clarified. And the reviewer also suggest that HepG2 cell line may be more suitable for apoptosis assay in this section.

4. In all in vivo experiments, NAFLD Activity Score (i.e., steatosis, lobular inflammation, and ballooning) for histological analysis in each NASH model should be included.

5. As mentioned above, due to the absence of steatosis and ballooning data, the reviewer generally observed no obvious change in the steatosis and ballooning based on Masson, Sirius Red, and Oil red O staining in Figure 3, 4, 6 & 7, and Extended Data Figure 3, 4, 6 & 7. Also, mice with Ripk1(D138N/D138N) expression or AAV-injected UGDH overexpression exhibit significantly reduced fibrosis. Does this mean that RIPK1 as a substrate of UGDH only affects the fibrotic process, but not inflammation and lipid metabolism?

(1) If yes, liver and serum TG, TC and NEFA contents and proinflammatory cytokines e.g., TNF- α , IL-6 and IL-1 β should be performed in each NASH model.

(2) Of note, previous reports (DOI: <https://doi.org/10.1038/s41418-020-00668-w>; DOI:<https://doi.org/10.1016/j.jhep.2019.11.008>) also have confirmed that RIPK1 suppression markedly decrease hepatic inflammation and liver lipid accumulation in response to metabolic stresses-related NAFLD or NASH progression. Thus, if the authors' results are not consistent with these findings, rational explanation need to be performed in this work.

(3) Additionally, the reviewer further observed that in authors' previous work (Nature Communications (2022) 13:7153, DOI:<https://doi.org/10.1038/s41467-022-34993-0>), they also investigate the function of Ripk1(D138N/D138N) in NAFLD development in response to HFD challenge. Although the authors use different dietary models in two different studies, they consistently confers protective effects in suppression of NAFLD/NASH progression via Ripk1(D138N/D138N); and identify Ripk1(D138N/D138N) as a key downstream regulator in pathological process.

Confusingly, in their previous paper, Ripk1 with D138N/D138N mutant not only significantly reduced fibrosis but also inflammation and lipid deposition, as indicated by H&E staining, Oil red O staining and Masson staining. Since RIPK1 is the main downstream target of hepatocyte apoptosis, why the different results regarding inflammation, lipid metabolism and fibrosis could be happened? If it is simply due to dietary model differences, then the reviewer suggest that HFD-mediated NAFLD should be also examined in the current study. If it is due to upstream regulatory factors of RIPK1 (i.e., SENP1 or UGDH), these biological process is not in a RIPK1-dependent manner. This point should be significantly clarified.

6. Raw western blotting bands need to be included in files.

7. Clinical characteristics of patients used in this work should be included as tables. For healthy liver, why the patients agreed to take liver biopsy? This is a key ethical problem that should be clearly indicated in the manuscript.

8. Is the UGDH-RIPK1 axis specific to hepatocytes? Will they participate the activation of KC or HSCs?

Reviewer #3 (Remarks to the Author):

This is an interesting study highlighting for the first time a new molecular mechanisms underlying the control of hepatocyte apoptosis in NASH. Nevertheless, there are some issues that need to be addressed in order to support the overall message of this work.

Figure 1i. UGDH deficiency sensitized hepatocytes to TNF-induced RIPK1-dependent apoptosis. The authors studied the formation of complex II consisting of FADD, RIPK1 and caspase-8 which initiates the death signaling of TNF-induced RIPK1-dependent apoptosis. Blot showing RIPK1 expression is clearly missing on the Figure 1i (part IP : anti-FADD). Could the authors add this blot?

Extended Data Figure 1e,f,g,h. Since the decreased protein expression of UGDH in the liver of CD-HFD or AMLN fed mouse was not regulated at transcriptional level, what would be the possible molecular mechanisms regulating its expression ?

Figure 2. Since the authors have access to liver tissues from patients with no steatosis, simple steatosis and NASH, could the authors characterized the expression of pS166 RIPK1 and CC3 in these different samples to reinforce the role of UGDH in NASH ? Regarding the decreased expression of UGDH in NASH patients. Did the authors look at the expression of mRNA ?

Figure 3f. At 8 weeks of feeding, the *Ugdh^{f/f}* mice fed CD-HFD still seems to have a lot of expression of the UGDH in comparison with mice fed CD-HFD for 12 weeks (Figure 2d). Could the authors add an experiment showing the decreased expression levels of UGDH in mouse liver according to the time feeding with CD-HFD ?

Figure 4d. Blot showing UGDH expression is missing on this figure, could the authors add this blot?

Figure 5c. To explore whether UDP-GlcA can inhibit RIPK1 activation in other models with RIPK1 activation, the authors studied the formation of complex I in BMDM upon stimulation with Flag-TNF and immunoprecipitation with anti-Flag antibody. In this complex I, leading to the activation of survival and inflammation pathways, RIPK1 is rather ubiquitinated and not phosphorylated on p-S166 (which is a marker of RIPK1 activation in the induction of RIPK1-dependent apoptosis or necroptosis). Revealing with an anti-pS166 RIPK1 antibody seems to shows a typical picture of protein ubiquitination and not phosphorylation (lack of band around 70 kDa). The authors must rather used an antibody anti-RIPK1 and an antibody anti-TRADD to better characterize the formation of complex I.

Figure 5h and Extended Data Figure 3. Two bands were detected on blot revealed with an anti-p-S166 RIPK1 while only one band is present on most of the other blots showing phosphorylation of RIPK1 on Ser166 in this paper ? On Figure 5h, there is a decrease in caspase 8 expression for hepatocyte *Ugdh^{f/f}* treated with T/C and siUxs1 whereas the tubulin signal is not modified ?

Figure 5i. Could the authors explain in the figure legend what does it mean L.E. p-S166 RIPK1. On this figure, UDP-Glc (1mM) can inhibit the phosphorylation of RIPK1-KD. Could the authors also show the effect of UDP-Glc on RIPK1-dependent apoptosis induced by T/5z7 in BMDM.

Figure 5m. in this in vitro kinase assay it seems that addition of UDP-GlcA was able to inhibit the low level of RIPK1-KD (D156A) phosphorylation observed on the blot ? Could the authors have a comment on this?

Extended Data Figure 5h. The authors showed that UDP-GlcA inhibited the formation of necrosome induced by T/S/Z treatment. It would be interesting to study the effect of UDP-GlcA on the formation of complex II in BMDM treated with T/5z7 (IP FADD and blot FADD, RIPK1, Caspase-8, and p-S166 RIPK1).

Figure 6a. Blot showing UGDH expression is missing on this figure, could the authors add this blot? Could the addition of UDP-GlcA affect the expression of UGDH ?

Figure 7c. Blot showing UGDH expression is missing on this figure, could the authors add this blot?
Could the extinction of Usx1 affect the expression of UGDH ?

In the paragraph Quantification and statistical analysis. The authors mentioned that all cell death data are presented as mean +/- SD of one representative experiment. Each experiment was repeated at least 3 times independently with similar results. Moreover in the figure legends, on several occasions regarding cell death measured as function of time by SytoxGreen positivity assay, it is indicated that data are represented as mean +/- SD of n=5 or n=3 independent samples of one experiment. This representation of the results for cell death experiments is problematic. The authors must average their three experiments to present the results and make statistical analyses.

Minor :

In the paragraph Immunoprecipitation page 32 : « For complex II immunoprecipitation, MEFs were seeded in 10 cm dishes ». In this paper, it was hepatocytes that have been used to study complex II formation.

In the paragraph Measurement of UDP-Glc and UDP-GlcA concentrations page 35 : « Methods for the extraction of intracellular nucleotide sugars and measurement of UDP-Glc and UDP-GlcA were described previously (Wang, et al., 2019) ». This reference is not included in the reference list.

Reviewer #1 (Remarks to the Author):

In this manuscript, the authors conducted a targeted siRNA screen using 114 siRNA pools for genes associated metabolic enzymes and identified UGDH as a suppressor of hepatocyte apoptosis induced by TNF α . In addition, they confirmed that UDP-GlcA catalyzed by UGDH directly binds to the kinase domain of RIPK1 to inhibit its activation and improve liver damage.

This research reveals the novel role of UGDH and UDP-GlcA in the pathogenesis of NASH. These results are interesting and clinically meaningful. While there were some additional questions the authors might consider addressing in this paper.

Reply: We thank this reviewer for the strong support! We have worked hard to address all of the comments made by this reviewer.

1. In fig.1, the results about hepatocyte apoptosis with Nec-1 treatment is confused. The authors claimed that UGDH deficiency promotes the activation of RIPK1 and converts RIA into RDA. Whether RIA or RDA is an apoptosis form induced by TNF α , why UGDH deficiency alone has no effect on apoptosis but has a promotion on apoptosis induced by TNF α ?

Reply: We thank the reviewer for raising insightful comments to guide us to revise the manuscript. Apoptosis can be activated when cell-surface death receptors like TNFR1 are engaged by their ligands like TNF α (the extrinsic apoptosis pathway). Our data show that UGDH deficiency alone did not induce apoptosis spontaneously but promotes apoptosis induced by TNF α in hepatocytes. This result is consistent to its role in regulating RIPK1 kinase activation, as RIPK1 usually acts downstream of death receptors, especially in TNF α signaling. In unstimulated condition, RIPK1 is unlikely to get activated and induce cell death even in the absence of its inhibitory proteins. Indeed, many key inhibitors of RIPK1 kinase, when depleted, promote apoptosis induced by TNF α rather than induce apoptosis in the absence of TNF α , such as cIAP1/2 (PMID:18485876), TAK1 (PMID:28842570), ABIN-1 (PMID:29203883), TBK1 (PMID:30146158; PMID:30420664), NEMO and IKK α/β (PMID: 26344099). Since UGDH is also an inhibitor of RIPK1 kinase activity, deletion of UGDH was able to promote TNF α -induced RIPK1 kinase-dependent apoptosis and UGDH deficiency alone is unlikely to induce RIPK1 kinase-dependent apoptosis in unstimulated condition.

If TNF α -induced apoptosis is RDA, the apoptosis of WT group with NEC-1S should be inhibited rather than unchanged as shown in the figure; If TNF α -induced apoptosis is RIA, so the apoptosis of the UGDH knockout group with NEC-1S should be returned to the same level as that of the control group rather than far lower than that shown in the figure.

Reply: We apologize for any confusion that has been made and not mentioning it clearly regarding RIA and RDA. Cell death, including apoptosis, is not the default response of TNF α stimulation and only proceeds when a protective brake, or cell-death checkpoint, is inactivated within the pathway. The outcome of TNF α stimulation is mainly regulated by the assembly of two protein complexes downstream of TNFR1 (PMID: 34813352). The membrane-bound Complex I forms within seconds of TNF α sensing and predominantly leads to MAPK- and NF- κ B-dependent gene activation. How this complex further evolves to induce cell death is less clear but involves the assembly of a secondary cytosolic complex, Complex II, which originates from the binding of FADD to TRADD and/or RIPK1. Complex II functions as a cytosolic platform for the binding and activation of Caspase-8, leading to the induction of extrinsic apoptosis. Two cell-death

checkpoints are known to counteract apoptosis induction by these pro-death complexes. First, the NF- κ B-dependent up-regulation of prosurvival genes that counteract Caspase-8 activation, such as cFLIP, protects cells from RIPK1 kinase-independent apoptosis (RIA). Second, suppression of RIPK1 kinase activity by inhibitory proteins, such as TAK1 (PMID:28842570), IKK α/β (PMID: 26344099), TBK1 (PMID:30146158; PMID:30420664), in Complex I prevents RIPK1 kinase-dependent apoptosis (RDA). CHX (Cycloheximide) is a translation inhibitor, which can suppress NF- κ B-dependent up-regulation of prosurvival genes in TNF α stimulation, and thus induce RIA. TNF α /CHX is commonly used as an apoptosis inducer, it only induces RIA in WT cells under normal condition (PMID:18485876; PMID:30146158; PMID:29203883). This is the same for WT hepatocytes, as our data show that RIPK1 kinase inhibitor, Nec-1s, was unable to inhibit TNF α /CHX-induced apoptosis in WT hepatocytes (**Fig. 1f-h** in the revised manuscript). However, TNF α /CHX-induced RIA can switch to RDA or even necroptosis (for example, PMID:29203883), when RIPK1 is prone to activation under certain conditions, such as brake for RIPK1 kinase activity is inactivated. For example, TBK1 is an inhibitory protein for RIPK1 kinase activity in Complex I of TNF α signaling, in TBK1-deficient MEF cells, TNF α /CHX induced more apoptosis than that in WT cells, which can be totally inhibited by Nec-1s, while Nec-1s did not inhibit TNF α /CHX-induced apoptosis in WT MEF cells (PMID:30146158, Fig. 2E-G). Since UGDH also inhibits RIPK1 kinase activation in TNF α signaling, deletion of UGDH promoted TNF α /CHX-induced apoptosis and converted RIA into RDA in response to TNF α . We have revised this section regarding RIA and RDA to avoid misleading.

2. Line 130-132, the results description should be corrected, as the data suggest necroptosis involved in the UGDH deficiency-induced cell death.

Reply: We fully agree with the reviewer that necroptosis might be involved in UGDH deficiency-induced cell death as UDP-GlcA was able to inhibit necroptosis induced by necroptotic trigger in BMDMs (**Fig. 5e** in the revised manuscript). Indeed, robust RIPK3 expression is essential for necroptosis, yet several publications indicate that hepatocytes do not express RIPK3 protein (PMID:36037995; PMID:26077809; PMID:27756058). A recent study revealed that RIPK3 is epigenetically silenced in both human and mouse hepatocytes, affecting their ability to undergo necroptosis (PMID:36037995). This is consistent to our data, as in the presence of pan-caspase inhibitor, zVAD, TNF α /CHX-induced hepatocytes cell death was inhibited (**Fig. 1f** in the revised manuscript), noting that TNF α /CHX/zVAD is a commonly used necroptosis stimuli in RIPK3-expressing cells (for example, PMID:26900751; PMID:29203883). In addition, inhibition of RIPK3 by its kinase inhibitor GSK'872 did not prevent TNF α /CHX-induced cell death in both WT and UGDH-deficient hepatocytes (**Fig. 1f**). Thus, necroptosis is not involved in UGDH deficiency-induced cell death at least in hepatocytes when stimulated with TNF α /CHX. We have revised this sentence to avoid misleading.

In fig.2A, the number of samples for UGDH quantitative analysis is inconsistent with that of western data.

Reply: We apologize for the confusion that has been made and not mentioning it clearly regarding the human liver samples. In Fig.2a, there are 6 samples for no steatosis control (#1, #2, #3, #13, #14, #24), we run the same sample (#1) in all three blots as a control for normalization and quantification. The number of samples for UGDH quantitative analysis is consistent with that of western data. We also added this information in the respective legend to avoid misleading. In addition, raw data for both blots and quantification are provided along with the revised manuscript as Source Data File.

3. In fig.3 about Ugdhf/f;Alb-Cre;Ripk1D138N/D138N, overexpression Ripk1D138N/D138N couldn't make phosphorylated RIPK1 in parent hepatocytes loss, which should be taken into consideration.

Reply: In **Fig.3b, 3d** and **3f**, RIPK1-D138N/D138N was able to substantially reduce the signal of activated RIPK1 as determined by p-S166 RIPK1 levels. In most cases, RIPK1-D138N/D138N was able to block p-S166 RIPK1 signal. However, there might be a very weak background signal, especially when performing immunofluorescence for p-S166 RIPK1 when using the p-S166 RIPK1 antibody. Indeed, background signal of p-S166 RIPK1 immunoblotting and immunofluorescence in RIPK1-D138N condition is commonly seen in many other cases (for example, PMID:30146158, Fig. 3F). In order to minimize the effect of background signal, we processed these images parallelly. The results show that p-S166 RIPK1 signal is substantially higher in UGDH-deficient liver section than that in WT liver section, and RIPK1-D138N/D138N can largely block these activated RIPK1 signal in UGDH-deficient liver sections.

4. In fig. 3b, compared with the lane 3, the expression of RIPK1 is significantly reduced. Does *ugdh* loss has an effect on RIPK1 expression? If so, what is the reason?

Reply: In Fig. 3b, compared with lane 3, RIPK1 level is decreased in lane 4 when cells were treated with TNF α /CHX. This is due to caspase-dependent cleavage of RIPK1 during apoptosis. RIPK1 is known to be cleaved during TNF α -induced apoptosis. This is commonly seen in many other cases when cells undergoing TNF α -induced apoptosis (for example, PMID:30146158, Fig. S3G, S3I, S3M, S3O). We did not observe any effect of UGDH loss on RIPK1 protein levels in untreated condition, for example, **Fig. 1h** (lane 4 vs lane 1), **Fig. 1i** (lane 4 vs lane 1), **Fig. 3b** (lane 1 vs lane 3).

5. In fig.5 & fig. 6, replenishment with UDP-GlcUA is difficult to understand. UDP-GlcUA transporters are located in the ER and not in outer cell membranes therefore there is no mechanism for its uptake into cells. How is the UDP-GlcA proposed to get into the cells when added to the media?

Reply: The solute carrier-35 D1 gene (*SLC35D1*) encodes a nucleotide sugar transporter in the endoplasmic reticulum that transports UDP-GlcUA, which are needed for the biosynthesis of chondroitin sulfate (PMID: 17952091). Indeed, the expression of *SLC35D1* is not restricted to endoplasmic reticulum, but also located on the plasma membrane of cells (PMID:31243371, ED fig. 2a). Thus, UDP-GlcA might be uptaken through *SLC35D1* that located in plasma membrane to get into the cells. This is consistent to our data as supplemented hepatocytes and BMDMs with UDP-GlcA was able to increase the cellular levels of UDP-GlcA in these two cell types (**Supplementary Fig. 6b, 6e** in the revised manuscript). We also added this information in the revised manuscript. We thank the reviewer for raising this insightful comment to guide us to revise the manuscript.

Reviewer #2 (Remarks to the Author):

In this work, the authors reported the role of UDP-glucose 6-dehydrogenase (UGDH) in NASH progression using a plethora of in vivo and in vitro rodent models. UGDH is identified as a key endogenous suppressor of RIPK1. UGDH directly interacts with RIPK3, and therefore blocks RIPK3 hyperactivation via changing UDP-glucose to UDP-glucuronate, which facilitates the UDP-glucuronate-associated inactivation of RIPK3 and balance hepatocytes metabolism homeostasis, further prevents development of CD-HFD-induced NASH phenotype.

Overall, massive amounts of data from in vivo and in vitro models of NASH have been provided to strongly support the important role of UGDH in the regulation of RIPK3 signaling and the pathogenesis of NASH. The concept of disturbance of liver metabolic homeostasis by UGDH-RIPK3 axis pathway is highly innovative. The manuscript is generally well-written, work flow is logical and quantity is impressive. The methodology employed appears to be appropriate and the experiments correctly controlled. However, many weak points in the current work need to be clarified and performed.

Reply: We thank this reviewer for the strong support! We have worked hard to address all of the comments made by this reviewer.

My specific comments about this manuscript are as follows:

1. In first part of result, the authors determine UGDH as a key suppressor of hepatocytes apoptosis in response to TNF- α administration. TNF- α -associated two apoptotic modalities, especially RIPK1, are accordingly highlighted. A number of previous studies have confirmed that necrosome RIPK1-RIPK3 core (DOI:<https://doi.org/10.1016/j.jhep.2019.11.008>; DOI:<https://doi.org/10.1016/j.cell.2018.03.032>) play an important role in NAFLD/NASH development. However, authors do not indicate how they determined that RIPK1 is the only substrate or key substrate for UGDH. Since RIPK1-RIPK3 may synergistically participate in cell apoptosis process, the role of UGDH in RIPK3 should be performed. The reviewer suggest the authors use mass spectrometry or multi-omics assay to determine (1) substrate specificity of UGDH-RIPK1 and (2) whether UGDH also functions on RIPK3.

Reply: We fully agree with the reviewer that RIPK1-RIPK3 may synergistically participate in cell apoptosis process. However, several publications indicate that hepatocytes do not express RIPK3 protein (PMID:36037995; PMID:26077809; PMID:27756058). A recent study revealed that RIPK3 is epigenetically silenced in both human and mouse hepatocytes (PMID:36037995). This is consistent to our data, as in the presence of pan-caspase inhibitor, zVAD, TNF α /CHX-induced hepatocytes cell death was inhibited (**Fig. 1f** in the revised manuscript), noting that TNF α /CHX/zVAD is a commonly used necroptosis stimuli in RIPK3-expressing cells (for example, PMID:26900751; PMID:29203883). In addition, we have shown that inhibition of RIPK3 by its kinase inhibitor GSK'872 did not prevent TNF α /CHX-induced cell death in both WT and UGDH-deficient hepatocytes (**Fig. 1f**). This data excluded the function of RIPK3 kinase in driving cell death in UGDH-deficient hepatocytes that stimulated by TNF α /CHX. Thus, it's unlikely that UGDH functions on RIPK3 in hepatocytes. We have highlighted this result in the revised manuscript to indicate the rational that focusing on RIPK1 but not RIPK3. We thank the reviewer for raising this insightful comment to guide us to revise the manuscript.

We found that UGDH suppresses RIPK1 activation in TNF α signaling through converting UDP-Glc into UDP-GlcA. We also proved that RIPK1 is a direct target for the metabolite UDP-GlcA (not UGDH). That saying, UGDH suppresses RIPK1 indirectly, thus it's not applicable to use mass spectrometry to

determine the UGDH-RIPK1 interaction in this study. Indeed, we revealed a new regulatory mechanism of RIPK1 kinase by UDP-GlcA directly in this study. We performed a thermal shift assay to determine the direct interaction between UDP-GlcA and RIPK1. The addition of UDP-GlcA to RIPK1 protein in the thermal shift assay increased its T_m by 7.2 °C, suggesting the binding of UDP-GlcA to RIPK1 (**Fig. 5j** in the revised manuscript).

2. In addition, Molecular modeling and docking method for UGDH and RIPK1 interaction assay is good one used in this work, this reviewer also prefer to observe UGDH-RIPK1 binding via protein GST pull-down assay, this detection is probably more intuitive and it also corroborates the molecular docking results.

Reply: We performed molecular docking for UDP-GlcA in binding with RIPK1 kinase domain in our study. Protein GST pull-down assay is not applicable here to determine the interaction between UDP-GlcA and RIPK1, as UDP-GlcA is a metabolite. Instead, we performed a thermal shift assay to determine the direct interaction between UDP-GlcA and RIPK1. The addition of UDP-GlcA to RIPK1-KD protein in the thermal shift assay increased its T_m by 7.2 °C, suggesting the direct binding of UDP-GlcA to RIPK1-KD (**Fig. 5j** in the revised manuscript).

3. In vitro experiments, primary hepatocytes and BMDMs are used to study apoptosis process, the topic of this work is to investigate role of UGDH in NASH progression by regulating RIPK1-associated hepatocyte apoptosis. Thus, liver-derived cell types may be a better choice for this study. The reviewer wonder know why the author used BMDMs instead of other liver-related cells (e.g., human HepG2 or rodent AML12). MEFs cells are used in authors' previous work (Nature Communications (2022) 13:7153, DOI:<https://doi.org/10.1038/s41467-022-34993-0>) to explore role of RIPK1 in NASH progression, but in this paper, BMDMs are used to establish genetic cells in vitro experiments. Are there any different reasons or phenomena between MEFs and BMDMs? This point need to be clarified. And the reviewer also suggest that HepG2 cell line may be more suitable for apoptosis assay in this section.

Reply: We apologize for the confusion that has been made and not mentioning it clearly regarding the cell lines chosen for studying RIPK1-related cell death. We fully agree with the reviewer that liver-derived cell type may be a better choice than BMDMs for NASH-related study. Indeed, we performed most of the experiments on primary hepatocytes, including studying the direct effect of UDP-GlcA on TNF α /CHX-induced apoptosis in UGDH-deficient hepatocytes (**Fig. 5a, 5b** in the revised manuscript). Since our study also revealed a new regulatory mechanism of RIPK1 kinase by the metabolite UDP-GlcA, and RIPK1 kinase is known to be central in RDA (RIPK1 kinase-dependent apoptosis) and necroptosis, it's reasonable to test whether UDP-GlcA also functions as a RIPK1 kinase inhibitor in a general manner in well-established RIPK1-related cell death models. Hepatocytes and hepatocyte-derived cell line cannot undergo necroptosis as they do not express RIPK3 (PMID:36037995). We also applied a commonly used RDA stimuli, TNF α plus TAK1 inhibitor 5Z-7-Oxozeaenol (TNF α /5z7, PMID:29891719) on WT hepatocytes. We found that TNF α /5z7 was unable to induce cell death in WT hepatocytes (**Fig. R1a**). Indeed, we screened the most used RDA and necroptosis stimuli (PMID:29078411), including TNF α /SM164 (RDA), TNF α /5z7 (RDA), TNF α /SM164/zVAD (necroptosis), TNF α /5z7/zVAD (necroptosis), and TNF α /CHX/zVAD (necroptosis), as well as the commonly used RIA (RIPK1 kinase-independent apoptosis) stimuli TNF α /CHX, on hepatocytes, and found that only TNF α /CHX can induce cell death on WT hepatocytes (**Fig. R1a**). Since TNF α /CHX induces only RIA in WT hepatocytes, our data show that UDP-GlcA did not inhibit cell death induced by TNF α /CHX in WT hepatocytes (**Fig. 5a** in the revised manuscript).

Most liver-derived cell types are cancer cell lines, such as HepG2. It's well-accepted that most cancer

cell lines (~83%) do not express RIPK3, which leads them to escape from necroptosis (PMID:30157175). Almost all liver cancer cell lines are resistant to necroptosis due to loss of RIPK3 (PMID:30157175, Fig. 2B). Thus, it's not applicable to use liver-derived cell lines to study the general function of UDP-GlcA on necroptosis. MEFs and BMDMs are both well-established cell types that can undergo RDA and necroptosis by respective stimuli, and they are commonly chosen as RIPK1-related cell death model cell types (For example, PMID:35544574; PMID: 29078411). To test whether UDP-GlcA has a general inhibitory function on RDA and necroptosis, we chose BMDMs and used TNF α /5z7 and TNF α /SM164/zVAD as RDA and necroptosis inducer, respectively. The reason for us to choose BMDMs but not MEFs here is that when we supplemented both cell lines with UDP-GlcA, BMDMs showed substantial increased cellular level of UDP-GlcA, suggesting the successful transport of UDP-GlcA from the medium into cells (**Supplementary Fig. 6e** in the revised manuscript), while MEFs seem to be unable to transport UDP-GlcA from the medium into cells (**Fig. R1b**). The reason for different efficiency of UDP-GlcA transport in BMDMs and MEFs might be attributed to the different expression levels of UDP-GlcA transporter, as transport of UDP-GlcA from medium into cells require the nucleotide sugar transporter (PMID: 17952091).

Together, (1) we have used UGDH-deficient primary hepatocytes to study the direct function of UDP-GlcA on RIPK1 activation and apoptosis to link the molecular mechanism to NASH progression. (2) We also studied whether UDP-GlcA is sufficient to inhibit RIPK1 activation in a general manner in other models with activated RIPK1, including TNF α /5z7-induced RDA and TNF α /SM164/zVAD-induced necroptosis in BMDMs, which is a commonly used cell type for studying RIPK1-related cell death and capable of transporting UDP-GlcA from the medium into cells. We thank the reviewer for pointing out this issue, we have clarified this point in the revised manuscript.

Figure R1. a, WT hepatocytes were pre-treated with CHX (1 μ M), 5z7 (0.5 μ M), SM164 (0.5 μ M) or zVAD (Z, 10 μ M) as indicated for 0.5 h followed by TNF α (10 ng/ml) for 24 h. Cell death was measured by SytoxGreen positivity assay. **b**, MEFs were supplemented with different concentration of UDP-GlcA for 12 h, intracellular UDP-GlcA concentrations were then determined.

4. In all in vivo experiments, NAFLD Activity Score (i.e., steatosis, lobular inflammation, and ballooning) for histological analysis in each NASH model should be included.

Reply: We thank the reviewer for this suggestion, NAS is the sum of separate scores for steatosis (0-3), hepatocellular ballooning (0-2), and lobular inflammation (0-3). We have now included NAFLD Activity Score (NAS) in each NASH model (**Fig. R2**, or **Supplementary Fig. 2f, 4d, 5e, 8d, 9c** in the revised manuscript).

Figure R2. NAFLD Activity Score (NAS) in each CD-HFD-induced NASH model. NAS is the sum of separate scores for steatosis (0-3), hepatocellular ballooning (0-2), and lobular inflammation (0-3).

5. As mentioned above, due to the absence of steatosis and ballooning data, the reviewer generally observed no obvious change in the steatosis and ballooning based on Masson, Sirius Red, and Oil red O staining in Figure 3, 4, 6 & 7, and Extended Data Figure 3, 4, 6 & 7. Also, mice with Ripk1(D138N/D138N) expression or AAV-injected UGDH overexpression exhibit significantly reduced fibrosis. Does this mean that RIPK1 as a substrate of UGDH only affects the fibrotic process, but not inflammation and lipid metabolism?

(1) If yes, liver and serum TG, TC and NEFA contents and proinflammatory cytokines e.g., TNF- α , IL-6 and IL-1 β should be performed in each NASH model.

Reply: We have shown hepatic TG levels in each NASH model in our manuscript. As kindly suggested by the reviewer, we further included hepatic levels of total cholesterol (TC) and nonesterified fatty acids (NEFA) in each NASH model in the revised manuscript (Fig. R3, or Supplementary Fig. 2g, 4e, 8e, 9d in the revised manuscript). We also included the levels of hepatic proinflammatory markers for each NASH model in the revised manuscript (Fig. R4, or Supplementary Fig. 2h, 4h, 8g, 9f in the revised manuscript). Our data suggest that UGDH deficiency did not alter hepatic steatosis and hepatic inflammation in CD-HFD-induced NASH model, and inactivating RIPK1 kinase in UGDH-deficient mice has no obvious effect on hepatic steatosis and inflammation as well.

Figure R3. Hepatic TG, TC and NEFA levels in each CD-HFD-induced NASH model.

Figure R4. Expression of hepatic proinflammatory markers in each CD-HFD-induced NASH model.

(2) Of note, previous reports (DOI: <https://doi.org/10.1038/s41418-020-00668-w>; DOI: <https://doi.org/10.1016/j.jhep.2019.11.008>) also have confirmed that RIPK1 suppression markedly decrease hepatic inflammation and liver lipid accumulation in response to metabolic stresses-related NAFLD or NASH progression. Thus, if the authors' results are not consistent with these findings, rational explanation need to be performed in this work.

Reply: We fully agree with the reviewer that RIPK1 activation has been shown to promote hepatic steatosis and hepatic inflammation in a conventional high-fat diet (HFD)-induced NAFLD model (PMID:33208891, PMID:31760070). However, our data indicate that in CD-HFD-fed UGDH-deficient mice, suppression of RIPK1 kinase was unable to reduce hepatic steatosis and inflammation. Indeed, HFD-induced NAFLD model displays less severe pathological outcomes than that of CD-HFD and human NASH, particularly the fibrogenesis process (PMID: 36263163, PMID:30449681). We thus compared the role of RIPK1 kinase in the pathogenesis of NAFLD in these two different NAFLD models. Consistent to previous reports, we found that 12-week CD-HFD feeding induced much higher levels of hepatic steatosis and hepatic inflammation than that of 16-week HFD feeding (**Fig. R5**). Consistent to the role of RIPK1 kinase in controlling hepatic steatosis and hepatic inflammation in HFD-induced NAFLD, we found that inhibition of RIPK1 kinase by D138N kinase-dead knockin mutation also decreased hepatic steatosis and proinflammatory markers expression as compared to ND-fed controls in this 16-week HFD-induced NAFLD model (**Fig. R5**). However, in 12-week CD-HFD-induced NASH model, we found that the effect of RIPK1 kinase in promoting hepatic steatosis and inflammation was largely compromised (**Fig. R5**). Thus, dietary model differences are involved in the effect of RIPK1 kinase in controlling NAFLD pathogenesis. Indeed, this can be also found in other cases. For example, AMPK re-activation can suppress hepatic steatosis in a HFD-induced NAFLD (Boudaba, et al., *EBioMedicine*, PMID:29343420), but failed to do so in CD-HFD-induced NASH model (Zhao, et al., *Science*, PMID:32029622, Fig. 4F, S7D)

Previously, we have also investigated whether UGDH-mediated RIPK1 suppression plays a role in the pathogenesis of HFD-induced NAFLD by feeding mice with a conventional HFD for 16 weeks to induce moderate hepatic steatosis and inflammation. We first characterized the expression levels of UGDH in HFD-fed livers. Unlike CD-HFD, HFD had minor effect on UGDH levels (**Fig. R6a**, or **Supplementary Fig. 5a** in the revised manuscript). Nonetheless, UGDH deficiency still promoted RIPK1 activation in HFD-fed livers (**Fig. R6b**, or **Supplementary Fig. 5b** in the revised manuscript). We also observed a significant increase of serum levels of ALT and AST, suggesting exaggerated liver damage, in HFD-fed UGDH-deficient mice compared with that of control mice, which is markedly reduced by RIPK1-D138N mutation (**Fig. R6c**, or **Supplementary Fig. 5c** in the revised manuscript). In keeping with this observation, HFD-fed UGDH-deficient livers showed increased number of TUNEL-positive cells, which can be decreased by

inhibition of RIPK1 kinase (**Fig. R6d**, or **Supplementary Fig. 5d** in the revised manuscript). In addition to a propensity for liver damage, HFD-fed UGDH-deficient mice, in comparison to *Ugdh^{f/f}* mice on the same diet, also exhibited greater levels of lipid accumulation and steatosis in the liver, as indicated by H&E and Oil red O staining (**Fig. R6e**, or **Supplementary Fig. 5e** in the revised manuscript), as well as by levels of hepatic TG and TC (**Fig. R6f**, or **Supplementary Fig. 5f** in the revised manuscript). Importantly, the hepatic steatosis and lipid accumulation in the liver of *Ugdh^{f/f};Alb-Cre;Ripk1^{D138N/D138N}* mice were considerably lower when compared to that of *Ugdh^{f/f};Alb-Cre* mice (**Fig. R6e, f**). Moreover, hepatocyte-specific UGDH ablation led to increased mRNA levels of multiple proinflammatory markers in mouse liver after 16-week HFD feeding (**Fig. R6g**, or **Supplementary Fig. 5g** in the revised manuscript). Notably, the hepatic expression of those proinflammatory markers in *Ugdh^{f/f};Alb-Cre;Ripk1^{D138N/D138N}* mice were also significantly lower than those in the *Ugdh^{f/f};Alb-Cre* mice (**Fig. R6g**). Those results suggested that in a conventional HFD model, which stimulates less severe NAFLD pathology than CD-HFD, UGDH-mediated RIPK1 suppression controls not only hepatocytes cell death and liver damage but also hepatic steatosis and inflammation.

Figure R5. WT and RIPK1-D138N mice (24-week-old) were fed with a HFD for 16 weeks or CD-HFD for 12 weeks. H&E staining and NAS score of liver sections of indicated mice (**a**). Oil red O staining of liver sections of indicated mice (**b**). Measurement of liver TG, TC and NEFA levels of indicated mice (**c**). Quantitative RT-PCR analysis of hepatic pro-inflammatory cytokines and chemokines of indicated mice (**d**). $n = 6$ mice for each genotype. Each dot represents an individual mouse. Mean \pm s.e.m. Unpaired two-tailed t-test.

Figure R6. a, Representative immunoblotting analysis of UGDH protein levels in the livers of 24-week-old mice fed ND or a conventional HFD for 16 weeks. $n = 3$ mice per group. **b**, Immunofluorescence images of p-S166 RIPK1 of liver sections from 24-week-old mice with indicated genotypes after feeding with a HFD for 16 weeks. Graph depicting numbers of p-S166 RIPK1⁺ cells on liver sections of indicated genotypes. $n = 6$ mice for each genotype. **c**, Serum levels of ALT and AST of 24-week-old mice with indicated genotypes after feeding with a HFD for 16 weeks. $n = 6$ mice for each genotype. **d**, Immunofluorescence images of TUNEL assay on liver sections from 24-week-old mice after feeding with ND or a HFD for 16 weeks. Graph depicting numbers of TUNEL⁺ cells on liver sections of indicated groups. $n = 6$ mice for each genotype. **e-g**, Mice (24-week-old) with indicated genotypes were fed with a HFD for 16 weeks. H&E and oil red O staining of liver sections of indicated mice (**e**). Measurement of liver TG and TC of indicated mice (**f**). Quantitative RT-PCR analysis of hepatic pro-inflammatory cytokines and chemokines of indicated mice (**g**). $n = 6$ mice for each genotype.

(3) Additionally, the reviewer further observed that in authors' previous work (Nature Communications (2022) 13:7153, DOI:<https://doi.org/10.1038/s41467-022-34993-0>), they also investigate the function of Ripk1(D138N/D138N) in NAFLD development in response to HFD challenge. Although the authors use different dietary models in two different studies, they consistently confers protective effects in suppression of NAFLD/NASH progression via Ripk1(D138N/D138N); and identify Ripk1(D138N/D138N) as a key

downstream regulator in pathological process. Confusingly, in their previous paper, Ripk1 with D138N/D138N mutant not only significantly reduced fibrosis but also inflammation and lipid deposition, as indicated by H&E staining, Oil red O staining and Masson staining. Since RIPK1 is the main downstream target of hepatocyte apoptosis, why the different results regarding inflammation, lipid metabolism and fibrosis could be happened? If it is simply due to dietary model differences, then the reviewer suggest that HFD-mediated NAFLD should be also examined in the current study. If it is due to upstream regulatory factors of RIPK1 (i.e., SENP1 or UGDH), these biological process is not in a RIPK1-dependent manner. This point should be significantly clarified.

Reply: As explained above (**Fig. R5 and R6**), our data suggest that dietary model differences are involved in the effect of RIPK1 kinase in controlling NAFLD pathogenesis. In a conventional HFD-induced NAFLD model, which stimulates moderate hepatic steatosis and inflammation, UGDH deficiency was able to promote hepatic steatosis and inflammation in a RIPK1 kinase-dependent manner, while in CD-HFD-induced NASH model, which induces much severer NAFLD pathology than HFD, hepatic steatosis and hepatic inflammation were way higher than that of HFD, the effect of RIPK1 kinase in driving hepatic steatosis and inflammation was largely compromised, and RIPK1 kinase only controls hepatocytes apoptosis and liver damage as well as liver fibrosis. Since UGDH was reduced in CD-HFD-fed liver, while HFD had minor effect on its expression, using CD-HFD-induced NASH model is more pathologically relevant. Nonetheless, we have included the above data and clarified the difference of RIPK1 kinase in controlling different pathology in different dietary models in the revised manuscript.

6.Raw western blotting bands need to be included in files.

Reply: We have included the raw blots along with the revised manuscript and provided them as a Source Data file.

7.Clinical characteristics of patients used in this work should be included as tables. For healthy liver, why the patients agreed to take liver biopsy? This is a key ethical problem that should be clearly indicated in the manuscript.

Reply: We thank the reviewer for this suggestion, we have included the clinical characteristics of patients in the revised manuscript as Supplementary Table. As indicated in the manuscripts, non-steatotic liver was used as the control. As clarified in the Method section that “steatotic livers were obtained from individuals with NAFLD or NASH who underwent liver biopsy or steatotic liver transplantation”, while “samples from nonsteatotic liver were collected from the normal donor livers”, not from liver biopsy. Of note, we also indicated in the Method section that “all donor livers were allocated via China Organ Transplant Response System from 2017 to 2021. The donors were enrolled in the study on a volunteer basis, and the families of organ donors were approached for consent. Written informed consent was obtained from subjects or families of all participants.”

8.Is the UGDH-RIPK1 axis specific to hepatocytes? Will they participate the activation of KC or HSCs?

Reply: The Human Protein Atlas (HPA) RNA-sequencing data showed that UGDH mRNA is highly enriched in hepatocytes and rarely expressed in KCs and HSCs (**Supplementary Fig. 1b** in the revised manuscript). Consistent to its mRNA expression levels in human, UGDH is also highly expressed in mouse hepatocytes, and rarely expressed in KCs and liver endothelial cells (**Supplementary Fig. 1d** in the revised manuscript). Due to the specific expression of UGDH in hepatocytes, UGDH-RIPK1 axis may be specific to hepatocytes.

Reviewer #3 (Remarks to the Author):

This is an interesting study highlighting for the first time a new molecular mechanisms underlying the control of hepatocyte apoptosis in NASH. Nevertheless, there are some issues that need to be addressed in order to support the overall message of this work.

Reply: We thank this reviewer for the strong support! We have worked hard to address all of the comments made by this reviewer.

Figure 1i. UGDH deficiency sensitized hepatocytes to TNF-induced RIPK1-dependent apoptosis. The authors studied the formation of complex II consisting of FADD, RIPK1 and caspase-8 which initiates the death signaling of TNF-induced RIPK1-dependent apoptosis. Blot showing RIPK1 expression is clearly missing on the Figure 1i (part IP : anti-FADD). Could the authors add this blot?

Reply: We thank the reviewer for pointing this issue, we have now included the RIPK1 blot in the IP part (**Fig. 1i** in the revised manuscript).

Extended Data Figure 1e,f,g,h. Since the decreased protein expression of UGDH in the liver of CD-HFD or AMLN fed mouse was not regulated at transcriptional level, what would be the possible molecular mechanisms regulating its expression?

Reply: Considering that UGDH protein, but not mRNA, levels were markedly decreased under conditions of NASH, we surmised that there was post-transcriptional regulation of UGDH during NASH pathologies, such as ubiquitination-dependent degradation. In line with this notion, UGDH have been found to have multiple ubiquitination sites (PhosphoSitePlus, <https://www.phosphosite.org/proteinAction.action?id=10632&showAllSites=true>). In addition, UGDH may also interact with multiple E3 ubiquitin protein ligase (BioGrid, <https://thebiogrid.org/113205/summary/homo-sapiens/ugdh.html>). To investigate the mechanism by which NASH reduces UGDH protein levels, we used palmitic acid (PA) to treat hepatocyte, which was performed to mimic *in vivo* NAFLD pathologies, to test whether we can observe UGDH reduction by PA in a cell-based model. However, we found that PA was unable to decrease UGDH levels (**Supplementary Fig. 1j** in the revised manuscript), which suggests that the decline of UGDH proteins levels in NASH is not caused by PA per se. Due to the lack of an appropriate cell-based model that can mimic NASH-induced UGDH reduction, it's challenging to elucidate the mechanism of NASH-induced UGDH reduction. We thank the reviewer for raising this excellent question, future study using *in vivo* NASH model to screen for upstream regulators of UGDH protein levels, such as E3 ubiquitin protein ligase, is needed. We have discussed this limitation of our study in the revised manuscript.

Figure 2. Since the authors have access to liver tissues from patients with no steatosis, simple steatosis and NASH, could the authors characterized the expression of pS166 RIPK1 and CC3 in these different samples to reinforce the role of UGDH in NASH? Regarding the decreased expression of UGDH in NASH patients. Did the authors look at the expression of mRNA?

Reply: We thank the reviewer for this suggestion, we have characterized the levels of pS166 RIPK1 and CC3 in no steatosis control and NASH samples (**Fig. R7a**, or **Supplementary Fig. 3f** in the revised manuscript). We also determined the mRNA levels of UGDH in NASH samples, we did not observe significant reduction of UGDH mRNA in NASH livers as compared to no steatosis control (**Fig. R7b**, or **Supplementary Fig. 1i** in the revised manuscript), which is consistent to that in the mouse NASH models.

Figure R7. a, Representative immunoblotting analysis of p-S166 RIPK1 and CC3 protein levels in human NASH livers and no steatosis controls. $n = 3$ individuals per group. **b,** Quantitative RT-PCR analysis of UGDH mRNA levels in livers from human NASH patients and no steatosis controls. $n = 3$ individuals per group.

Figure 3f. At 8 weeks of feeding, the *Ugdh*^{h/f} mice fed CD-HFD still seems to have a lot of expression of the UGDH in comparison with mice fed CD-HFD for 12 weeks (Figure 2d). Could the authors add an experiment showing the decreased expression levels of UGDH in mouse liver according to the time feeding with CD-HFD?

Reply: We thank the reviewer for this suggestion, we have included the blot showing UGDH expression in samples with ND-fed or CD-HFD-fed for 8 weeks and 12 weeks, respectively. We found that 8 weeks of CD-HFD feeding slightly reduced UGDH expression in the liver (**Fig. R8**, or **Supplementary Fig. 1g** in the revised manuscript), suggesting that UGDH is reduced in a NASH severity-dependent manner.

Sup. Fig. 1g

Figure R8. Representative immunoblotting analysis of UGDH protein levels in the livers of 20-week-old mice fed ND or CD-HFD for indicated time periods. $n = 3$ mice per group.

Figure 4d. Blot showing UGDH expression is missing on this figure, could the authors add this blot?

Reply: We thank the reviewer for this suggestion, we have included this blot (**Fig. 4d** in the revised manuscript).

Figure 5c. To explore whether UDP-GlcA can inhibit RIPK1 activation in other models with RIPK1 activation, the authors studied the formation of complex I in BMDM upon stimulation with Flag-TNF and immunoprecipitation with anti-Flag antibody. In this complex I, leading to the activation of survival and inflammation pathways, RIPK1 is rather ubiquitinated and not phosphorylated on p-S166 (which is a marker of RIPK1 activation in the induction of RIPK1-dependent apoptosis or necroptosis). Revealing with an anti-pS166 RIPK1 antibody seems to show a typical picture of protein ubiquitination and not phosphorylation (lack of band around 70 kDa). The authors must rather use an antibody anti-RIPK1 and an antibody anti-

TRADD to better characterize the formation of complex I.

Reply: We thank the reviewer for this suggestion, we have now included total RIPK1 blot and TRADD blot in complex I in this experiment (**Fig. 5c** in the revised manuscript). We fully agree with the reviewer that Flag-TNF α stimulation alone leads to RIPK1 ubiquitination in complex I and thus the activation of survival and inflammation pathways. Although Flag-TNF α stimulation does not induce lethal RIPK1 activation, a nonlethal pool of RIPK1 is still activated in complex I (for example, PMID:27819682, Fig. 2c, lane 2, 3, and ED Fig. 5a, b, lane 2; PMID:30146158, Fig. 3F lane 3; PMID:36414671, Fig. 4a, lane 2, 3, and Fig. 4b, lane 2). Since RIPK1 is mainly in a ubiquitinated form in complex I, this nonlethal pool of activated RIPK1 in complex I is also in a ubiquitination state, which can be prevented by RIPK1 kinase inhibition (**Fig. R9**, or Newton et al., *Nature*, PMID:27819682, ED Fig. 5b). In this study, we found that UDP-GlcA was able to inhibit RIPK1 kinase, thus we also observed inhibition of RIPK1 activation but not its ubiquitination by UDP-GlcA in complex I in BMDMs that were stimulated by Flag-TNF α (**Fig. 5c** in the revised manuscript).

Newton et al., *Nature*, ED Fig. 5b

Figure R9. Data from Newton et al. show that Flag-TNF α was able to induce RIPK1 activation as well as ubiquitination in complex I in WT BMDMs. This pool of activated RIPK1 is also in a ubiquitination state. Of note, RIPK1-D138N kinase dead was able to abolish this pool of RIPK1 activation without affecting its ubiquitination.

Figure 5h and Extended Data Figure 3. Two bands were detected on blot revealed with an anti-p-S166 RIPK1 while only one band is present on most of the other blots showing phosphorylation of RIPK1 on Ser166 in this paper?

Reply: Activated RIPK1 seems to have additional post-translational modifications in addition to S166 phosphorylation, which may lead to upshift of p-S166 RIPK1 band on SDS-PAGE, for example, PMID:27819682, ED Fig. 3g (**Fig. R10a**). However, probably due to technical differences, sometimes this upshift of p-S166 RIPK1 band may not present on SDS-PAGE in different experiments, for example, PMID:27819682, Fig. 2a (**Fig. R10b**). This can be found in many other cases (Dziedec et al., *Nat Cell Biol*, PMID:29203883, Fig. 2; Lafont et al., *Nat Cell Biol*, PMID:30420664, Fig. 4; Amin et al., *PNAS*, PMID:29891719, Fig. 5). Thus, the reason for sometimes two bands of p-S166 RIPK1 was observed, but sometimes only one band was observed in this study, may also attribute to technical differences among different SDS-PAGEs, especially with regarding to band shift.

Figure R10. Data from Newton et al. show that activated RIPK1 may show two bands on an SDS-PAGE (a), but sometimes only show one band in other SDS-PAGE (b) probably due to technical difference among different SDS-PAGEs, especially with regarding to band shift.

On Figure 5h, there is a decrease in caspase 8 expression for hepatocyte Ugdhf/f treated with T/C and siUxs1 whereas the tubulin signal is not modified?

Reply: We thank the reviewer for pointing out this issue. We have repeated this blot, and found that UXS1 knockdown does not affect the expression of caspase 8 under T/C treatment (Fig. 5h in the revised manuscript). We noticed that the decrease in caspase 8 expression in this lane is due to film developing issues (probably a bubble on this lane). We apologize for the confusion that has been made due to our technical issues regarding this blot.

Figure 5i. Could the authors explain in the figure legend what does it mean L.E. p-S166 RIPK1. On this figure, UDP-Glc (1mM) can inhibit the phosphorylation of RIPK1-KD. Could the authors also show the effect of UDP-Glc on RIPK1-dependent apoptosis induced by T/5z7 in BMDM.

Reply: We thank the reviewer for this suggestion. L.E. means long-time exposure of the film. We have indicated it in the legend of this figure. As kindly suggested by the reviewer, we further investigated the effect of UDP-Glc on RIPK1-dependent apoptosis induced by T/5z7 in BMDMs. Similar to T/S/Z-induced necroptosis (Supplementary Fig. 7c in the revised manuscript), we found that UDP-Glc also had very minor effect on T/5z7-induced apoptosis in BMDMs (Fig. R11, or Supplementary Fig. 7b in the revised manuscript), which is consistent to the minor effect of UDP-Glc on RIPK1 activation.

Figure R11. BMDMs were pre-treated with 5z7 (100 nM) for 0.5 h followed by TNF α (1 ng/ml) for indicated time in the presence or absence of UDP-Glc with indicated concentration. Cell death was measured as a

function of time by SytoxGreen positivity assay.

Figure 5m. in this in vitro kinase assay it seems that addition of UDP-GlcA was able to inhibit the low level of RIPK1-KD (D156A) phosphorylation observed on the blot? Could the authors have a comment on this?

Reply: We thank the reviewer for this suggestion. Since UDP-GlcA interacts with multiple sites of RIPK1-KD in addition to D156, UDP-GlcA was able to further inhibit RIPK1-D156A phosphorylation, suggesting that the interaction of UDP-GlcA and RIPK1-D156 mainly but not totally contribute to the interaction of UDP-GlcA and RIPK1-KD. We have included this comment in the revised manuscript.

Extended Data Figure 5h. The authors showed that UDP-GlcA inhibited the formation of necrosome induced by T/S/Z treatment. It would be interesting to study the effect of UDP-GlcA on the formation of complex II in BMDM treated with T/5z7 (IP FADD and blot FADD, RIPK1, Caspase-8, and p-S166 RIPK1).

Reply: We thank the reviewer for this suggestion. We have included the blots of complex II formation in BMDMs treated with T/5z7 (**Fig. R12**, or **Supplementary Fig. 6g** in the revised manuscript). The results suggest that UDP-GlcA inhibited the formation of complex II in T/5z7-treated BMDMs, which is consistent to the role of UDP-GlcA in suppressing RIPK1 activation and T/5z7-induced apoptosis in BMDMs.

Figure R12. BMDMs were pre-treated with 5z7 (100 nM) for 0.5 h followed by TNF α (1 ng/ml) for indicated time in the presence or absence of UDP-GlcA (0.2 mM). The complex II was isolated by immunoprecipitation of FADD, RIPK1 and Caspase 8 binding was revealed by immunoblotting.

Figure 6a. Blot showing UGDH expression is missing on this figure, could the authors add this blot? Could the addition of UDP-GlcA affect the expression of UGDH?

Reply: We thank the reviewer for this suggestion, we have included UGDH blot in this experiment (**Fig. 6a** in the revised manuscript). The result indicates that addition of UDP-GlcA did not affect the expression of UGDH, which is consistent to the data in Fig. 5b (lane 4).

Figure 7c. Blot showing UGDH expression is missing on this figure, could the authors add this blot? Could the extinction of Usx1 affect the expression of UGDH?

Reply: We thank the reviewer for this suggestion, we have included UGDH blot in this experiment (**Fig. 7c** in the revised manuscript). The result indicates that deletion of UXS1 did not affect the expression of UGDH, which is consistent to the data in Fig. 5h (lane 3, 4).

In the paragraph Quantification and statistical analysis. The authors mentioned that all cell death data are presented as mean +/- SD of one representative experiment. Each experiment was repeated at least 3 times independently with similar results. Moreover in the figure legends, on several occasions regarding cell death measured as function of time by SytoxGreen positivity assay, it is indicated that data are represented as mean +/- SD of n=5 or n=3 independent samples of one experiment. This representation of the results for cell death experiments is problematic. The authors must average their three experiments to present the results and make statistical analyses.

Reply: We thank the reviewer for this suggestion, we have now included average cell death data from all three independent experiments, and presented them as mean +/- s.e.m. of $n = 3$ independent experiments and made statistical analyses for all cell death experiments, including **Fig. 1b, 1d, 1f, 1g, 1j; 3a, 3c; 5a, 5d, 5e, 5g** and **Supplementary Fig. 4a; 6c; 7b, 7c, 7e, 7f**. These information as well as statistical method used for each cell death experiment have been included in the respective legend.

Minor :

In the paragraph Immunoprecipitation page 32 : « For complex II immunoprecipitation, MEFs were seeded in 10 cm dishes ». In this paper, it was hepatocytes that have been used to study complex II formation.

Reply: We thank the reviewer for pointing this mistake. Since we also performed complex II IP in BMDMs (**Supplementary Fig. 6g** in the revised manuscript), we have revised this sentence as “For complex II immunoprecipitation, hepatocytes or BMDMs were seeded in 10 cm dishes”.

In the paragraph Measurement of UDP-Glc and UDP-GlcA concentrations page 35 : « Methods for the extraction of intracellular nucleotide sugars and measurement of UDP-Glc and UDP-GlcA were described previously (Wang, et al., 2019) ». This reference is not included in the reference list.

Reply: We thank the reviewer for pointing this mistake. We have included this reference (PMID: 31243371) in the reference list of our revised manuscript.

REVIEWER COMMENTS

Reviewer #1 (Remarks to the Author):

The authors have answered some of my queries and added additional data for clarification. But there are two very important issues not to resolve.

1.The authors claimed that they did not observe any effect of UGDH loss on RIPK1 protein levels in untreated condition, but from fig3b, lane 1 compared with lane3, internal control band seems lighter in UGDH KO group, while RIPK1 band seems much darker. the data suggest that UGDH KO upregulated RIPK1 expression.

2.The authors' answer about UDPGA replenishment was not correctly. The reference (PMID:31243371, ED fig. 2a) cited by the authors couldn't support their point. In this reference, Dr. Weiwei Yang holds that UDP-GA couldn't get into cells directly and transfected SLC35D1 plasmids to make sure that cells can uptake the exogenous UDPGA. Although the authors give that UDPGA in cells increase after UDPGA complement, I am still surprised and puzzled that exogenous sugar can compensate for the lack of intracellular UDP-GA production. UDPGA is very hydrophilic and doesn't have known transporters on the plasma membrane (as opposed to the SCL35 transporters on the ER membrane).

Reviewer #2 (Remarks to the Author):

Authors have addressed all my questions. This manuscript is ready for the next step.

Reviewer #3 (Remarks to the Author):

I thank the authors to have carefully and clearly answered all my questions and requests. The paper is now ready for publication.

Reviewer #1 (Remarks to the Author):

The authors have answered some of my queries and added additional data for clarification. But there are two very important issues not to resolve.

Reply: We thank the reviewer very much for the supportive comments. We also appreciate the constructive comments from the reviewer to help us to further strengthen our manuscript.

1.The authors claimed that they did not observe any effect of UGDH loss on RIPK1 protein levels in untreated condition, but from fig3b, lane 1 compared with lane3, internal control band seems lighter in UGDH KO group, while RIPK1 band seems much darker. the data suggest that UGDH KO upregulated RIPK1 expression.

Reply: We thank the reviewer for pointing out this issue. We have repeated this blot together with its internal control, and further confirmed that UGDH knockdown does not affect the expression of RIPK1 (**Fig. 3b** in the revised manuscript). We noticed that the decrease in RIPK1 expression in lane 1 is due to film developing issues (probably a bubble on this lane). We apologize for the confusion that has been made due to our technical issues regarding this blot.

2.The authors' answer about UDP-GlcA replenishment was not correctly. The reference (PMID:31243371, ED fig. 2a) cited by the authors couldn't support their point. In this reference, Dr. Weiwei Yang holds that UDP-GA couldn't get into cells directly and transfected SLC35D1 plasmids to make sure that cells can uptake the exogenous UDP-GlcA.

Reply: We apologize for any confusion that has been made and not mentioning it clearly in response to the reviewer's previous comments with respect to UDP-GlcA transport. We are appreciated of the chance the reviewer has provided to clarify it. We fully agree with the reviewer that "In this reference (PMID:31243371), Wang et al. holds that UDP-GlcA couldn't get into cells directly and transfected SLC35D1 plasmids to make sure that cells can uptake the exogenous UDP-GlcA." Their results indicate that **1. the expression of SLC35D1 is not restricted to the ER membrane, and it also presents on the plasma membrane, 2. plasma membrane-expressed SLC35D1 is able to transport exogenous UDP-GlcA from the medium into cells.** However, the expression of SLC35D1 is highly tissue and cell type-specific. For example, the Human Protein Atlas (HPA) RNA-sequencing data showed that SLC35D1 mRNA is highly enriched in the liver (**Fig. R1a**) and mainly expressed by hepatocytes (**Fig. R1b**). Thus, SLC35D1 overexpression may be required in cells expressing low levels of SLC35D1 for transporting exogenous UDP-GlcA.

Figure R1. SLC35D1 is highly enriched in liver and highly expressed by hepatocytes. The Human Protein Atlas (HPA) RNA-sequencing data of UGDH expression in different tissues (a), hepatocytes and non-parenchymal liver cells (b).

Although the authors give that UDP-GlcA in cells increase after UDP-GlcA complement, I am still surprised and puzzled that exogenous sugar can compensate for the lack of intracellular UDP-GlcA production. UDP-GlcA is very hydrophilic and doesn't have known transporters on the plasma membrane (as opposed to the SCL35 transporters on the ER membrane).

Reply: We fully agree with the reviewer that UDP-GlcA is very hydrophilic, whose transport into cells must require a plasma membrane-expressed transporter. It's not clear whether there is a specific plasma membrane-expressed transporter for UDP-GlcA other than SLC35D1. Since SLC35D1 can transport UDP-GlcA (PMID:11322953, PMID:31243371), and its expression is not restricted to the ER membrane, but also on the plasma membrane (PMID:31243371), it is possible that plasma membrane-expressed SLC35D1 contributes to the transport of exogenous UDP-GlcA. Consistent with this notion, overexpression of SLC35D1 in cells expressing low levels of SLC35D1 was able to increase intracellular UDP-GlcA after UDP-GlcA supplement (PMID:31243371). However, the expression of SLC35D1 is tissue and cell-type

specific. That's saying, cells with high expression level of SLC35D1 may be able to transport exogenous UDP-GlcA. Indeed, SLC35D1 is highly enriched in the liver and mainly expressed by hepatocytes (**Fig. R1**).

In this study, we supplemented UDP-GlcA to hepatocytes and mice, and determined the effect of UDP-GlcA on hepatocytes cell death and liver pathology of NASH. Given the high expression of SLC35D1 in hepatocytes, it is possible that hepatocytes transport exogenous UDP-GlcA through SLC35D1. The successful transport of exogenous UDP-GlcA in hepatocytes was evidenced by the increased cellular level of UDP-GlcA after UDP-GlcA supplement in this study. In addition to hepatocytes, we also supplemented UDP-GlcA to immortalized BMDMs and MEFs, two cell types that are commonly used for RIPK1-dependent cell death study. We found that UDP-GlcA only inhibits RIPK1-dependent cell death in iBMDMs but not iMEFs (this is the reason that we choose iBMDMs but not iMEFs to study the role of UDP-GlcA in RIPK1-dependent cell death). Accordingly, UDP-GlcA supplement only increased cellular level of UDP-GlcA in iBMDMs but not iMEFs (**Fig. R2a**). We then examined the expression levels of SLC35D1 in hepatocytes, iBMDMs and iMEFs. We were able to detect SLC35D1 protein expression in both hepatocytes and iBMDMs, but rarely in iMEFs (**Fig. R2b**). Thus, cells with high expression of SLC35D1 may be capable of transporting exogenous UDP-GlcA, such as hepatocytes and iBMDMs.

Figure R2. a, MEFs were supplemented with different concentration of UDP-GlcA for 12 h, intracellular UDP-GlcA concentrations were then determined. **b**, Immunoblotting analysis of SLC35D1 protein expression in hepatocytes, BMDMs and MEFs.

Reviewer #2 (Remarks to the Author):

Authors have addressed all my questions. This manuscript is ready for the next step.

Reply: We thank the reviewer for the strong support of our work, and also for raising insightful comments to guide us to revise the manuscript.

Reviewer #3 (Remarks to the Author):

I thank the authors to have carefully and clearly answered all my questions and requests. The paper is now ready for publication.

Reply: We thank the reviewer for the strong support of our work, and also for raising insightful comments to guide us to revise the manuscript.

REVIEWERS' COMMENTS

Reviewer #1 (Remarks to the Author):

The authors have addressed all my issues.

Reviewer #1 (Remarks to the Author):

The authors have addressed all my issues.

Reply: We thank the reviewer for the strong support of our work, and also for raising insightful comments to guide us to revise the manuscript.